# The Development Analysis of the Romanian Traditional Product Market Based on the Performance Model for Sustainable Economic Development

**Silvius Stanciu [1], Monica Laura Zlati [2], Valentin Marian Antohi [3, *] and Cezar Ionut Bichescu [1]**

[1]  Center for Technology Transfer, "Dunarea de Jos" University of Galati, 800001 Galati, Romania; sstanciu@ugal.ro (S.S.); cezar.bichescu@ugal.ro (C.I.B.)

[2]  Department of Accounting, Audit and Finance, Stefan cel Mare University, 720229 Suceava, Romania; sorici.monica@usm.ro

[3]  Departament of Business Administration, Dunarea de Jos University/Faculty of Economics and Business Administration, 800001 Galati, Romania

*  Correspondence: valentin_antohi@yahoo.com

**Abstract:** This study aims at quantifying the degree of concentration of the traditional product market in Romania, and the sector's productivity and the economic performance of the different categories of traditional products are assessed. This can highlight a correlation between the dynamics of traditional products and the regional development of the relevant markets in Romania. The second aim of this study is to analyze the relationship between the economic profitability of the traditional products and the sector's sustainability. The third aim evaluates the relationship between the capital accumulations of specific companies and the evolution of their current assets. The information selected for the application was prospective (literature review, market observations, query, and data consolidation) and analytic revised (database analysis, hypothesis fixation, model conceptualization, model hypothesis testing, and conclusions to be drawn). The selected data were processed, aiming at developing a model for the sustainable development of the traditional products. The research information was collected based on the official registrations carried out between 2014 and 2018 by at the Ministry of Agriculture and Rural Development in Romania. During this period, Romania developed the National Traditional Product Registry (NTPR), which comprises a database of 647 traditional products. The assessment of economic performance was achieved through calculated performance test by a new proposed statistical model, named ZML. ZML suggests a market concentration analysis as an alternative to the Gini Struck method. The impact of the research consisted of an evaluation of the economic performance of traditional Romanian products in sustainable development terms.

**Keywords:** sustainable development; local initiative; economic performance; economic opportunities; competitive market; traditional product; statistical performance model

## 1. Introduction

The international economic context regarding trade with traditional agricultural products is clearly influenced by the market globalization phenomenon. Production and trade of traditional products represent a competitive challenge for small producers and a priority of European policy. An increase in the share of small agri-food producers may represent a balance factor and a response to the globalization of the Romanian market. The modern challenges of sustainable agriculture are

mainly related to the quality of exogenous factors (i.e., the quality of soil, the surface or ground water, and the air). Agriculture, an essential branch of the international economy, has undergone continuous expansion over the last decade. Extensive development has produced irreversible effects in terms of climate change, soil erosion, and biodiversity loss. By practicing intensive farming, the degradation of these resources has generated a chain of adverse effects [1]. Particularly, the use of fertilizers and greenhouse gas emissions has generated significant climate change, with a direct impact on the quality of life and food. New technologies aim at optimizing the agricultural production based on intensive strategies rather than an extensive development. However, there are researchers who support the idea of the negative impact of the excessive type of agriculture associated mainly with the production of raw materials [2,3].

Organic production, reducing consumption and losses, and producing traditional goods represent innovative guidelines for both new food production and a response to growth-based systems. A traditional product is based on a traditional recipe (the recipe refers to a specific way of production and/or processing and to a traditional technological process). These characteristics distinguish traditional products from similar products belonging to the same category [4].

In a competitive market economy, the agricultural market develops specific mechanisms, with extremely volatile behaviors. Based on the nature of the agro-food production, defined as inflexible in terms of price, the significant changes of production volume require significant periods of time to restore market balance [5,6].

Lately, the agriculture based on an efficient marketing process has become an important profit generator for farmers and the national economy. A study by Majeed et al. analyzed the effects of various factors on the profitability of agricultural economic entities [7]. Experts have identified evidence in favor of associating agricultural indicators, such as the production period and the conversion cycle, with the performance of the agricultural company [8]. The transformed traditional Romanian agricultural model under the impact of European Union (EU) agricultural policies is more productive, especially due to the financial opportunities provided by EU funding programs for the agricultural sector [9]. The evolution of achieving sustainable development objectives in Romania compared to the EU average shows that, despite recorded delays, Romania can reduce the gaps in the sustainable development domain [10]. Environmental crisis trigger factors are limited in the context of implementing an action model involving the following: rational resource management, the implementation of recycling technologies, pro-active economic processes in relation to environmental policy, the support and development of green industries, and other measures regarding sustainable development [11].

Romania has made progress related to the transformation of the national food system(s). Related to the classification made by the Global Food Security Index (GFSI) 2018, Romania comes in at the 38th position with a total score of 68.9 points, from a total number of 113 monitored countries. The Global Food Security Index is monitoring three strategic criteria: affordability, availability, and quality & safety.

The best score on all three criteria realized by Romania for food quality and safety is 32nd in the world [12], with a score of 72.6 points, similar to Hungary with 72 points and the Czech Republic with 73.7 points. Topping this classification are Portugal with 87.3 and France with 86.5. Based on food availability, Romania has the most favorable score from all three criteria, 67.5 points, compared to the first two positions: Singapore with 94.3 points and Qatar with 92.9 points. The dynamics trend between 2012 and 2018 was positive, and the net medium growth on all three criteria was 1.8 points. The criteria based on which countries are classified upon the dynamics analysis of food systems and the global effects of environmental changes. The GFSI is the first instrument of food safety examination from the point of view of accessibility, availability, and food quality in 113 countries. The GFSI mentions the involvement of Romania in increasing food standards, creating a guide for a well-balanced diet, a food improvement strategy, and data collection related to the deficiencies caused by nutrition. Romania food safety is sustained by the existence of a sufficient capacity to store food, according to the indicator. With all the improvements that the GFSI underlined related to

availability and accessibility, the GFSI report shows that food quality and safety has generally gone down, mostly due to poorly diversified diets and poor protein diets [12].

From the recognized perspective of cultural realities, the concept of a place of origin can be a source of added value for traditional agri-food products. The cultural fund and the positive capacity of transmitting local historical information generate significant influence on the international market transaction [13].

The European Union made consistent progress in protecting its traditional brands. The European recognition of traditional foods (on a certification process of origin) is protected under the Protected Geographical Indication (PGI). The Traditional Specialty Guaranteed (TSG) is part of the European Commission's E door website [14]. From this perspective, the European agri-food policy aims at protecting the names of specific products in order to promote their unique related characteristics to geographical origin by promoting their traditional features.

The European Union's financial measures support rural sustainable development areas as well as the agricultural production on the efficiency principles of agricultural policy [15]. In many situations, traditional producers are affected by a lack of financial means, although the level of investment is low compared to industrial food producers. However, Bolek mentioned that the existence of larger funds for traditional food producers leads to lower risks but also to a lower profit rate [16].

In this context, the encouragement of traditional producers has become a policy promoted at the European level as a means of supporting small producers and the sustainable development of rural areas from which traditional products originate. EU Regulation 1151/2012 on agricultural product and food quality schemes is designed for the European market specifically [17]. The measures relate to an increase in the quality of food (including traditional ones) while maintaining a diversity of agricultural and food production as well as fair competition. Quality is an indicator that decisively influences the growth of independence with respect to those who sell. The price, no longer seen as an independent variable, indicates buyer pressure influence [18]. This situation generates the need for agricultural and food products with identifiable characteristics, particularly in terms of tradition and geographical origin [19].

From the consumer's point of view, traditional production without the application of modern principles of food safety and quality can affect its health. Consumer behavior has become more refined, and the need for access to information has increased. Traditional products must be hygienically labeled and marketed accordingly. If there is no evidence of compliance with quality measures, the behavior of the food consumer can become selective due to a lack of confidence in ensuring food security and nutritional product concerns [20].

Studies show that education can have a positive impact on a consumer's behavior and the quality of food consumed [21]. Social food-related messages can be useful, taking into account the fact that it has been discovered that being familiar with these messages has improved the citizens' perceptions concerning what healthy food is, enhancing their acceptance and understanding [22].

Some researchers [23] have reconsidered findings in the traditional foods sector through a multidimensional sustainable development aims. Thus, their research defined some potential impact factors:

- the cultural factors that are closely related to the markets and their regional diversity;
- the homogenization of food production depending on the type of region and the synthesis of information regarding the nutritional benefits that are closely related to the cultural preferences of the population residing in a specific area. (As for traditional products, the potential affects factors that are emphasized by the diminished food biodiversity. This aspect is countered by a culture of traditional food consumption in a specific area.)

Boncinelli [24] makes a connection between the traditional product market and the dynamics of the need for brand name products (fast food products). He concludes that making people aware of the consumption of healthy foods is a crucial factor in protecting the traditional products within an area. The study exemplifies the extra virgin oil consumption in Italy. The results of the study show

that, within a heterogeneous type of market, the preferences determine the consumption both horizontally and vertically and enable the difference in, selection of, products to be consumed (the consumer's behavior). Detailed analysis of some traditional products has revealed the existence of a heterogeneous preference market in spite of the fact that this particular type of food is part of traditional diets.

Brunori [25] initiated a study based on the impact of localization within the chain of distribution connected to POD (Protected Origin Designation) and the PGI (Protected Geographical Indicator), which was stipulated even in the European Union's prescriptions on good practices of the traditional trading. Experts have shown that the perception based on position differs for the same type of product. It is in itself a factor that is sensitive to the local population's features.

Cacciolati [26] designed a study based on an analysis of several traditional products. It revealed that traditional products' distinct labeling and positioning have a major impact on a consumer's preference.

Campbell [27] analyzed the selling system of traditional products based on their food quality and security flaws. He emphasized the fact, in certain communities, the consumption risk of processed goods in households may lead to serious consequences regarding public safety. In this respect, it is highly suggested that both the producers and their products within the distribution chain be monitored by having in view first and foremost their main feature (the expiry date).

Coelho [28] developed a study on the improvement of certain POD cheese brands. He demonstrated that it is of utmost importance that small businesses or small factories manufacture traditional products to adopt management and control techniques for the collected raw materials. Thus, they should improve the quality of the products offered for consumption and the consumer's trust by leading to accordingly added value to the POD.

Defrancesco et al. [29] analyzed features that have a significant impact on health by taking into account several studies in the specialty field. They measured the quality of traditional products from the point of view of the consumer's willingness to pay (WTP). They reached the conclusion that the features of the environment and health increase accordingly. Based on their findings, a significant aspect of WTP refers to the product's specific taste, which makes a difference when it comes to purchasing two almost similar traditional products.

Rudawska [30] analyzed the consumer's loyalty when it comes to the consumption of traditional products. He concluded that the option to buy traditional products represents a consequence of growth in regional wealth and of growth in regional cultural perception and social identity. The quality curve exceeds that of quantity when it comes to consumers' options.

Lafuente [31] analyzed the innovation in the field of traditional products. He concluded that understanding the management process, as well as ensuring strategic resources and actions, in order to organize the ins and outs of the latest products in a portfolio is of utmost importance. Thus, the package of traditional products that a company launches at a certain date becomes sustainable only if the innovation strategies are viable and solid.

Pilone et al. [32] analyzed development policies of the consumer's acceptance in terms of traditional products. He presented a case study on the sustainable development related to environment factors and the PGI. The authors argued that traditional products in connection with innovation are a main competitive element. The certification of products leads to an increase in consumers' trust and WTP (consumer's willingness to pay) by creating a synergy for the food politics geared towards innovation, quality insurance, and sustainable development.

Barska and Wojciechowska-Solis [33] conducted a study that shows that there is a relationship between the gender of the respondents and the traditional food consumption models (the organoleptic qualities, the products' enriched quality, the consumers' curiosity, and the taste of new products). The choice of consuming traditional products is influenced by consumers' income, their education, and their awareness of the tangible and intangible attributes of these products.

In order to identify the consumers' profiles and create different promotion strategies for local and traditional products, Vlontzos [34] identified six major factors: the consumer's behavior, the incertitude regarding their health issues, costs, the influence of the media and friends, and

positioning in the store. The findings of the study showed that young adults were geared towards the consumption of local and traditional products, whereas the cost factor had a variable/different influence on the target group. They are also unsure about the influence of traditional products on the population's health.

In Romania, the legislation of traditional products has been an interesting development. Thus, until 2014, the lack of clear regulation of the field allowed for the national registration and certification of over 4000 foods as traditional products, many of which were industrially manufactured without respecting the principles of traditionality [19]. Having in view that the traditional food production regulations have proved to be permissive and unclear, in 2014 the Romanian Ministry of Agriculture and Rural Development developed new criteria for registering traditions that were focused on protecting consumers against abusive practices by imposing measures to ensure the correctness of traditional production [4]. During this period, Romania developed the National Traditional Product Registry (NTPR) in Romania, a database of 647 traditional products. The national system for traditional foods is intended to be a preliminary stage of certification at the European level.

Following the new regulations, the number of traditional products certified at the national level has significantly reduced. Therefore, the NTPR (with 647 products registered at the end of 2018), was revised [35]. As the consumer's behavior has become more refined and the access to information and the impact of technological developments have increased, the niche of healthy products (also known as bioproducts) has grown, too. Romania is known, on a national European and international level, as a reliable, trustworthy supplier in terms of bioproducts, especially for honey and honey-related products, cheeses, and meats. Among some other well-known Romanian brands are certain alcoholic drinks (Bihor palinka) and traditional preserves (pickles and vegetable stew), Romania making a significant effort to keep its national brands. This has lead to the development of a rural, national, and European heritage, because the brand is key for a company's contributing to a long-term relationship between the consumer and the producer [36, 37]. The European community has made substantial progress in protecting its traditional brands by recognizing protected origin products with geographical indications and guaranteed traditional specialties. Quality brands ensure brand identity and uniqueness through separate labeling systems for the European Union.

The purpose of this study is to quantify the degree of concentration of the traditional product market in Romania, as well as to assess the sector's productivity and the economic performance of the different categories of traditional products.

Based on a new statistical model, called ZML, this article analyzes (i) the relationship between the economic profitability of traditional products and the sustainability of the sector and (ii) the relationship between the capital accumulations of specific companies and the evolution of their current assets.

The rest of the article is organized as follows. Section 2 presents the data taken from the National Traditional Product Registry; Section 3 describes the research methodology of our study including the data sources, the means used, and the details of the performed analysis. In this section, we suggest a new statistical model (ZML) that quantifies the degree of concentration in the areas of productivity and economic performance of traditional products. Sections 4 and 5 present the outcome of the study; the research ends with a conclusion and observations regarding limitations.

## 2. Data Presentation from the National Traditional Product Registry

In the present study, we focus on the productive potential assessment of the Romanian traditional product market. This is defined by:

- specific characteristics regarding the connection to Romanian culinary traditions;
- the historical evolution of the products in the historical context of the region in which they are produced;
- the quality of products generated by the use of ecological raw materials obtained by means of traditional and local methods.

The context of the traditional market is reinforced by certain external factors:

a. the promotion of production at the European level through measures that encourage and finance traditional manufacturers;

b. the creation of promotion centers (events, fairs, and promotion exhibitions) for this type of product;

c. the integration of these traditional products into agro-tourism, ecotourism, and rural tourism.

The European Council endorsed a legislative package aimed to establish the principles and general requirements of food legislation [38]. This legislative package creates the prerequisites for food safety regulations and ensures the diversity of agro-food production in the context of fair competition.

This set of regulations created the necessary premises for the implementation of systems that certify the original character and quality of products in the context of making rural economies more efficient through increasing the production of traditional products and ensuring mechanisms for food quality standards, including the creation of the European Food Safety Authority.

In this context, a resolution of guaranteed traditional specialties of agricultural products and foodstuffs was adopted at a national level with the help of government. Based on these legislative measures adopted at the European and national level, premises for sustainable development were created, and Good Practice Guides for the certification of traditional products were published. These guides aim to define legal framework by conceptualizing Tender Specifications, establishing procedures for the certification of traditional products by regulating the marketing procedures related to traditional products and, importantly, by putting into place a set of measures to control the quality of products through specific bodies.

These steps led to the establishment of the NTPR, which, during the period 2014–2018, recorded information on a number of 647 traditional products belonging to 7 categories of products [35].

A total of 187 companies geographically distributed in 35 counties in Romania are registered in the national statistics of traditional food products. The data are centralized in Table 1.

**Table 1.** The geographical distribution of certified traditional products and the product categories to which they belong.

| County | Product Categories | Products | County | Product Categories | Products |
|---|---|---|---|---|---|
| Alba | 5 | 53 | Hunedoara | 4 | 11 |
| Arad | 3 | 3 | Iasi | 3 | 12 |
| Arges | 4 | 37 | Ilfov | 2 | 13 |
| Bacau | 1 | 1 | Maramures | 6 | 65 |
| Bihor | 2 | 6 | Mures | 1 | 1 |
| Bistrita Nasaud | 2 | 2 | Neamt | 2 | 36 |
| Botosani | 5 | 29 | Olt | 1 | 2 |
| Brasov | 5 | 175 | Prahova | 2 | 5 |
| Braila | 1 | 1 | Satu Mare | 5 | 27 |
| Bucuresti | 1 | 4 | SALAJ | 3 | 13 |
| Buzau | 2 | 24 | Sibiu | 2 | 11 |
| Caras Severin | 1 | 3 | Suceava | 4 | 18 |
| Cluj | 4 | 8 | Timis | 1 | 4 |
| Constanta | 1 | 1 | Tulcea | 3 | 19 |
| Covasna | 3 | 25 | Vaslui | 1 | 2 |
| Dambovita | 2 | 9 | Valcea | 3 | 12 |
| Galati | 1 | 7 | Vrancea | 1 | 5 |
| | | | Harghita | 1 | 3 |
| Statistics | | | | | |
| Total | 88 | 647 | of which unique item | 7 | 647 |
| Average | 2.5 | 18.5 | Std distribution | 1.48 | 30.73 |
| First quartile | 1 | 3 | Third quartile | 3.5 | 21.5 |

Source: Calculations carried out by the authors based on the information taken from [35].

Table 1 shows that, at a county level (NUTS III), the average number of product categories for which traditional product certification rights are requested is 2.5, whereas the average number of products is 18.5.

The vast majority of the traditional products were founded in Brasov (175), Maramures (65), and Alba County (53). The counties with the worst recordings of national products in NTPR are Bacau, Brăila, Mureş, Constanţa (a single product), Bistriţa Năsăud, Olt, and Vaslui (two traditional products).

The data presented reflects an uneven distribution of interest in the production, promotion, and marketing of traditional products in Romanian counties. Some traditional areas have maintained their leadership positions on the national market for traditional products (Figure 1).

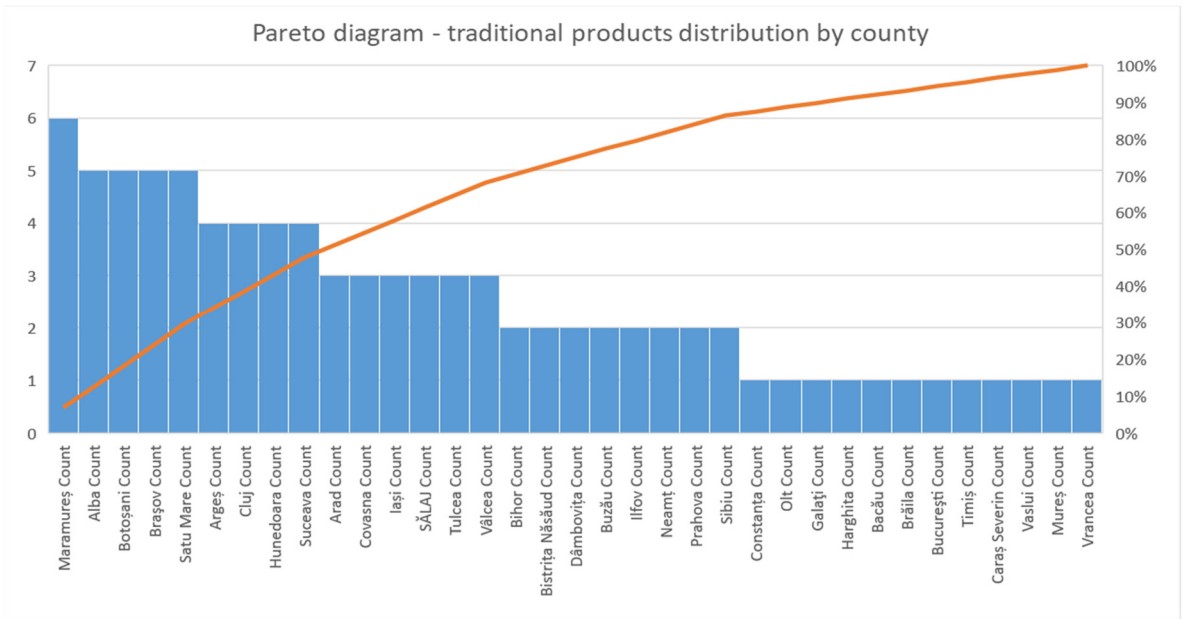

**Figure 1.** The Pareto traditional product distribution chart according to NUTS III classification.

The chart reflects the uneven distribution of traditional food production in Romania and areas with a high potential for it. From the point of view of the number of producers, according to Table 2, they act predominantly at county level (178 cases) and, to a small extent, at national level (22 cases). The food categories that traditional producers target are meat and meat products (51 producers and 241 certified products), milk and milk products (48 producers and 141 products for), and bread, bakery, and pastry (58 producers and 108 certified products). The average number of producers is about 30, and the average number of products is 99 (Table 2).

**Table 2.** Distribution of certified traditional products and their producers by product categories.

| List of Categories | Producer of the Certified Traditional Product | The Certified Traditional Product |
|---|---|---|
| Other | 2 | 2 |
| Beverages | 8 | 21 |
| Meat and meat products | 51 | 241 |
| Milk and milk products | 48 | 141 |
| Vegetables, fruit | 33 | 107 |
| Bread, bakery, and pastry | 58 | 108 |
| Fish | 9 | 27 |
| Statistics | | |
| Total | 209 | 647 |
| of which unique item | 187 | 647 |
| Average | 29.86 | 92.43 |
| Std distribution | 21.60 | 77.87 |
| First quartile | 8.5 | 24 |
| Third quartile | 49.5 | 124.5 |

Source: Calculations carried out by the authors, relying on the information taken from [35].

The best-represented products, from the point of view of the diversity of the product range, analyzed based on the product/producer share, are the meat products (4.7), fruits and vegetables

(3.24), and fish products (3.3). The lowest represented are bakery and pastry products (1.86), while for beverages the ratio is 2.62 products per producer. The overall average range is 3.45 products per manufacturer (Figure 2).

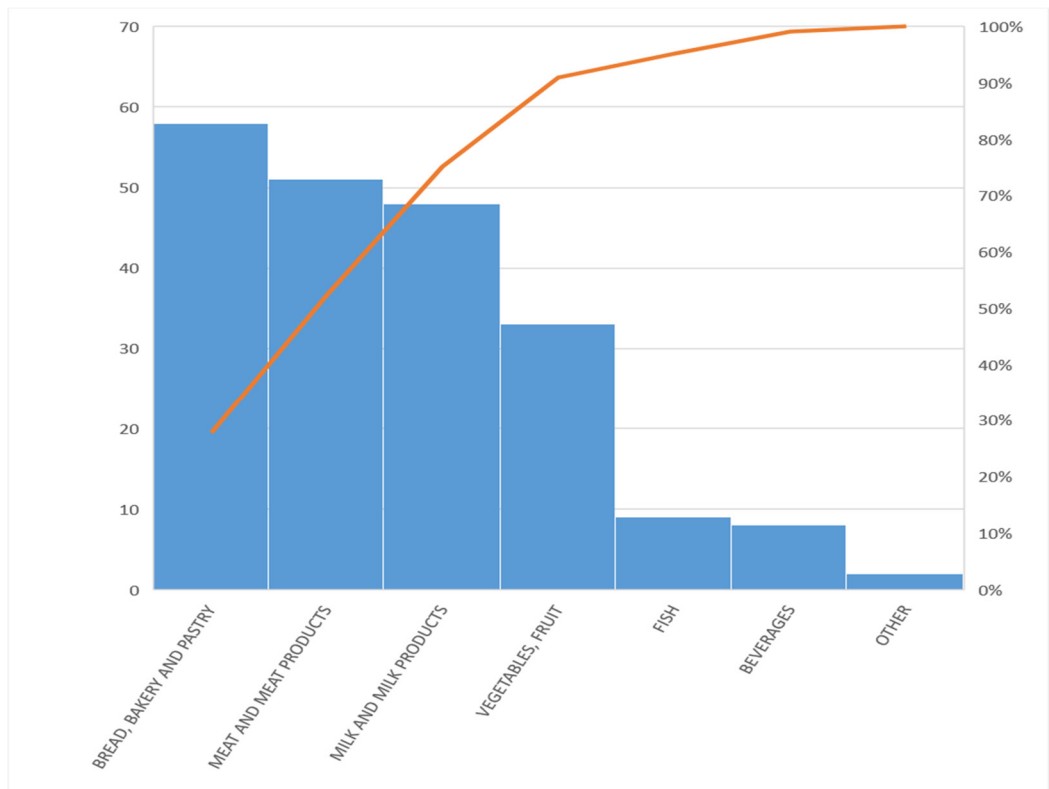

**Figure 2.** The Pareto traditional product and producer distribution chart by product categories.

Figure 3 shows the most developed categories of traditional products compared to those at an early stage (which are specific to relatively restricted areas or subject to marketing restrictions due to the potential health risk to the population, such as spirits). The evolution of the number of traditional products in the period 2014–2018 is shown in Table 3.

**Table 3.** The distribution over time of certified traditional products divided by product category.

| Total | 2014 | | 2015 | | 2016 | | 2017 | | 2018 | | Max | Min |
|---|---|---|---|---|---|---|---|---|---|---|---|---|
| | Other | 0 | Other | 2 | Other | 0 | Other | 0 | Other | 0 | Max | Min |
| | Beverages | 5 | Beverages | 15 | X | 0 | Beverages | 1 | X | 0 | 2015 | 2016 |
| 241 | Meat and meat products | 117 | Meat and meat products | 79 | Meat and meat products | 25 | Meat and meat products | 9 | Meat and meat products | 11 | 2014 | 2017 |
| 141 | Milk and milk products | 78 | Milk and milk products | 46 | Milk and milk products | 1 | Milk and milk products | 11 | Milk and milk products | 5 | 2014 | 2016 |
| 107 | Vegetables, fruit | 44 | Vegetables, fruit | 13 | Vegetables, fruit | 21 | Vegetables, fruit | 10 | Vegetables, fruit | 19 | 2014 | 2017 |
| 108 | Bread, bakery and pastry | 37 | Bread, bakery and pastry | 41 | Bread, bakery and pastry | 6 | Bread, bakery and pastry | 5 | Bread, bakery and pastry | 19 | 2015 | 2017 |
| 27 | Fish | 15 | Fish | 7 | Fish | 2 | Fish | 0 | Fish | 3 | 2014 | 2017 |
| 647 | **Total 2014** | 296 | **Total 2015** | 203 | **Total 2016** | 55 | **Total 2017** | 36 | **Total 2018** | 57 | 2014 | 2017 |
| **Max** | Meat and meat products | 2014 | Meat and meat products | 2015 | Meat and meat products | 2016 | Milk and milk products | 2017 | Vegetables, fruit | 2018 | | |
| **Min** | Other | | Other | | Other | | Other | | Other | | | |

Source: Author's calculations by relying on the information from [35].

Analyzing the data presented in Table 3, one can notice the fact that the 2014–2015 period was the most active in terms of the certification of traditional products due to the creation of the NRNT based on Order 724/2013. This certifies manufacturers that are present in the market of traditional products.

In two years' time, there were over 296 certified products belonging to six product categories. Once most traditional manufacturers certified their products, the process of certification normalized to an average of 50 products per year. The worst year in terms of certifications was the year 2016, when Romania underwent profound changes in the Fiscal Code regarding the taxation of authorized natural persons and legal entities, having an important negative influence on the traditional food product market (Table 4).

**Table 4.** The distribution of certified products and product categories by years in the National Traditional Product Registry (NTPR).

| Year | Products | Categories |
|------|----------|------------|
| 2014 | 296 | 6 |
| 2015 | 203 | 7 |
| 2016 | 55 | 5 |
| 2017 | 36 | 5 |
| 2018 | 57 | 5 |
| Total | 647 | 7 |

Source: Author's calculations based on the information from [35].

As for product categories, the producers' interest focused in particular on meat, milk, vegetable products, and bread and pastries. The last place in the ranking oscillates between fish and beverage products. Particular attention should be paid to fishery products, as the interest was stimulated by financing projects and structural funds to promote fishing and aquaculture (Figure 3).

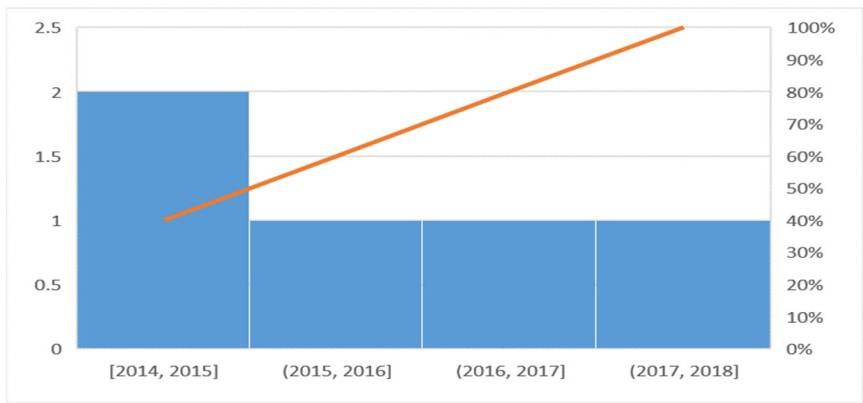

**Figure 3.** The Pareto distribution chart for certified products and product categories by years; source: [35].

The Pareto frequency chart reflects the trend mentioned above in terms of dynamics within the traditional product categories.

## 3. Material and Method

In order to analyze the sustainable economic development concentration degree based on the statistical performance score in the Romanian traditional product sector, working hypotheses were established as follows (study hypothesis):

**H1.** The dynamics of the traditional products certified in the Romanian economy is closely related to the regional development. The indicator is characterized by the regional macroeconomic context in relation to the cultural and historical values of the region. In addition, it has a yearly growth rate that was established during the last period of analysis. The indicator is in relation with the duration of the implementation of the NTPR [35].

**H2.** The number of traditional product categories is kept in a dynamic balance, and it varies in terms of product saturation. The applied criteria refer to the product categories and economic opportunities that are generated by promotion campaigns of traditional products.

**H3.** The profitability and cost-efficiency of certain traditional product categories is even greater as the number of certified traditional products increases. The sustainability of the branch development is due to the stabilization in the time of production.

**H4.** The evolution of a company's equity (segmented by category of traditional products) is in close connection with the evolution of the current assets as well as with the increase in profitability. The condition is satisfied if the profitability increase is higher than the increase in turnover.

**H5.** The need to take the branch to a more high-tech level involves the accumulation of debts in a relationship that is directly proportional to it. It considerably diminishes the capital accretive at the entity level. However, on a long-term basis, it is a condition for sustainable growth.

In order to carry out the study, the working scheme presented in Figure 4 was taken into account. The data were obtained using the NTPR records, available on the Ministry of Agriculture and Rural Development of Romania website. The Ministry of Public Finance in Romania collected the financial data. No questionnaires or interview methods were used.

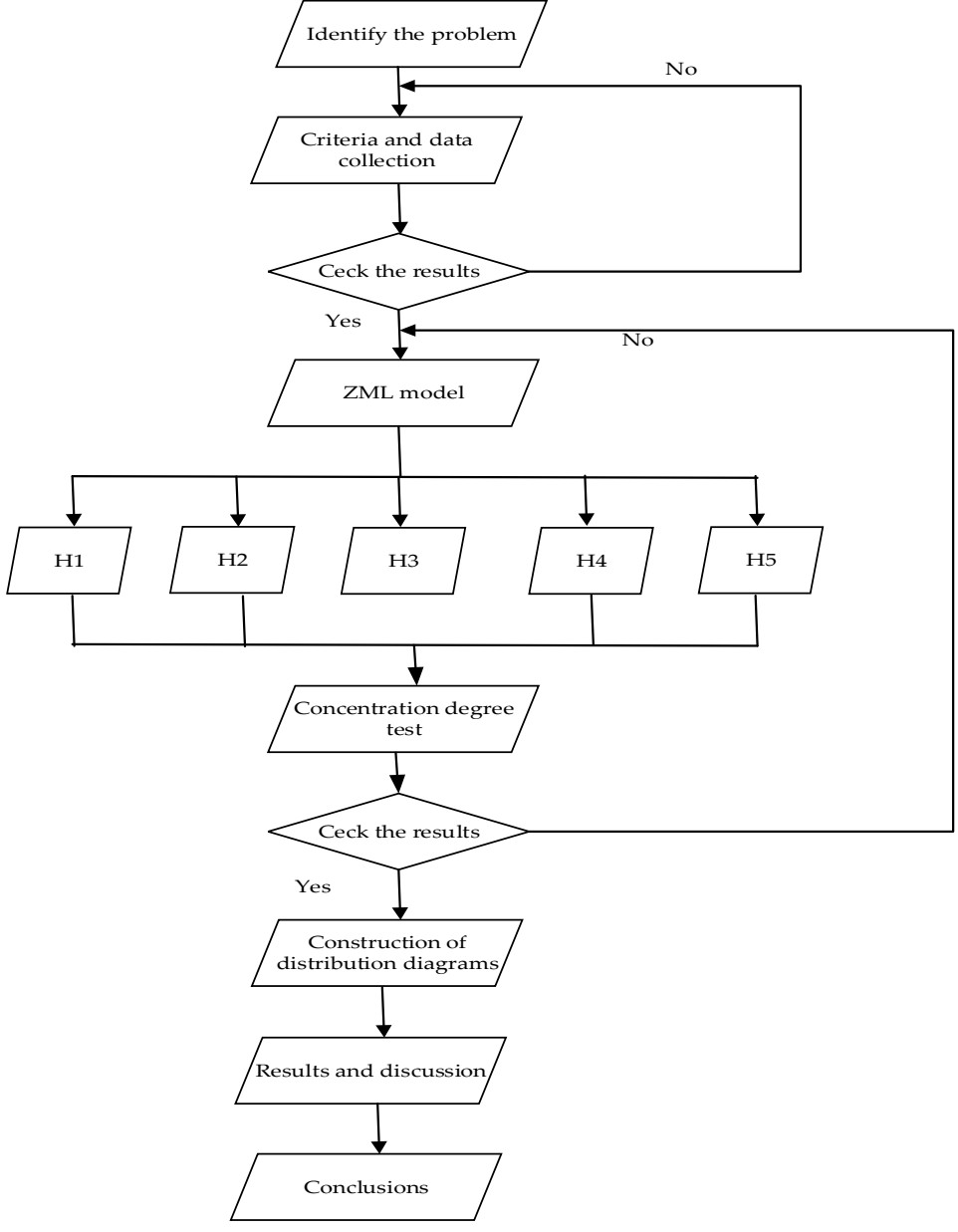

**Figure 4.** The methodology of the study.

The method consisted in analyzing the profitability of the dynamics and capital accumulation of economic agents selling different types of traditional products. The aim of the research was to achieve

an image of the sustainable development of the traditional product sector marketed in Romania by product categories. The methods used were the synthesis of the legislative framework and the study of the opportunities offered by the new directions of sustainable development in the field. Analysis of the accreditation bodies and financial analysis of the subsidiary were carried out. For this purpose, a new statistical model, called ZML, was specifically designed to quantify the degree of concentration regarding traditional products productivity and economic performance.

In the scientific literature, a close-up study of the degree of concentration, based on the Gini Struck method, is defined by Equation (1).

$$GSI = \sqrt{\frac{n\sum gi^2 - 1}{n-1}}. \tag{1}$$

where n is the number of factors, and gi is the specific weight of each factor of the series [19].

The limitations of the Gini Struck model are the relative distribution of the frequency of the series studied within the sample population and the lack of quantification of the impact of this distribution's general sample. We think that the new model developed by the researcher Zlati Monica Laura, generically named ZML, is more complex in quantifying the degree of concentration in the areas of productivity and economic performance compared to the model of analysis of the traditional product concentration on Romanian market traditional products [19].

The ZML model is defined as follows:

Let defined $Z^+ \in N^+$, the set consisting of a population that meets certain performance criteria $\alpha_i$ as follows:

- $\alpha_i$ represents the performance criterion that responds to the equation $\alpha_i - \alpha_{statistic} > 0$;
- $\alpha_{statistic}$ represents the performance limit of the criterion or the minimum value for which the criterion fulfills the performance condition.

The hypotheses of the proposed model are as follows:

**Hypothesis I.** We believe that ($\exists$) $i_k$ and $j_m$ so that ($\forall$) $i_k \in j_m$, ($\forall$) $j_m \in Z^*$, $k \in [1,p]$ and $m \in [1,r]$, where

- $p$—the total population included in the sample that meets the performance criteria;
- $m$—the number of categories comprised by the sample.

**Hypothesis II.** Then ($\exists$) $i_k^* \geq (\forall)$ ($i_k$) si $j_m^*(i_k) \geq (\forall)$ $j_k$ ($i_k$) so that the function of the concentration degree on the peak of the performance score is valid under the following conditions:

$$ZML^* = \frac{\sum_{m=1}^{r} j_m}{\sqrt{\frac{\sum_{m=1}^{r} j_m \cdot \left[1 - \left(\frac{i_k}{\sum_{k=0}^{p} i_k}\right)^2\right]}{(\sum_{m=1}^{r} j_m) - 1}}} > 0 \tag{2}$$

and

$$\begin{cases} \lim_{i \to \infty}(|\max(i_k) - ZML^*|) = \lim_{i \to \infty}(|i_k^* - ZML^*|) \to 0 \\ ZML^* \ll \sum_{k=1}^{p} i_k \\ ZML^* \to \sum_{k=1}^{p} i_k \Leftrightarrow i \to 1 \end{cases}.$$

**Hypothesis III.** Then ($\exists$) $i_k^{**} \leq (\forall)$ ($i_k$) and $j_m^{**}(i_k) \leq (\forall)$ $j_m$ ($i_k$) so that the function of the concentration degree on the lower limit of the performance model (ZML$^{**}$) is valid under the following conditions:

$$ZML^{**} = \frac{\sum_{m=1}^{r} j_m}{\sqrt{\frac{\sum_{m=1}^{r} j_m \cdot \left[\left|\frac{i_k}{\sum_{k=0}^{p} i_k}\right|^2 - 1\right]}{(\sum_{m=1}^{r} j_m) - 1}}} > 0 \tag{3}$$

and

$$\begin{cases} \lim_{i \to \infty} (|\min(i_k) - \text{ZML}^{**}|) = \lim_{i \to \infty} (|i_k^{**} - \text{ZML}^{**}|) \to 0 \\ \text{ZML}^{**} \ll \sum_{k=1}^{p} i_k \\ \text{ZML}^{**} \to \sum_{k=1}^{p} i_k \Leftrightarrow i \to 1 \end{cases}.$$

**Hypothesis IV.** In order to achieve the performance profile that any $i_k$ should adopt in view of gaining access to the leader's performance score and, implicitly, to maximize economic benefits, the difference

$$\Delta = \text{ZML}^{*} - \text{ZML}^{**} \tag{4}$$

is representative $\Leftrightarrow$ for $i_k$, $\alpha_{i_k} - \alpha_{statistic} = 0$.

The proposed model allows for the identification of the concentration level within the leader's area by means of the statistical model defined by Equation (1) as well as the position of the analyzed item ($i_k$) in the sample. A special emphasis should be made on the difference from the leader described in Equation (5).

A statistical model was developed, based on the least squares (WLS) model, for the demonstration of the formulated hypotheses. The model's equation is a cumulative one. It is in the form of $F = \sum \alpha_i * R_i + \varepsilon$, where

- $F$ stands for the dependent variable of the model;
- $\alpha_i$ stands for the regression coefficients;
- $R_i$ stands for the regressive variables;
- $E$ stands for the residual variable;
- $i$ stands for the number of the statistical observation.

As far as the data is concerned, the previously determined model presents itself as follows:

$$\text{Producers} = +0.202*\text{Products} + 0.0906*\text{Turnover} + 0.625*\text{GrossProfit}$$
$$(0.0992) \qquad (0.588) \qquad (0.313) \tag{5}$$

where n = 647. R-squared = 0.870 (standard errors in brackets).

The statistical tests used for verifying the ZML model applied the GRETL model (Equation (2)) and indicated the homology and the statistical significance of the ZML model at 87%, demonstrating its validity. Based on the Belsley–Kuh–Welsch test, the ZML model collinearity is not present. The *p*-value of the model for the F test is less than 0.2, which demonstrates that it is statistically significant (Table 5).

**Table 5.** Model 1: OLS, using remarks 1-647 (dependent variable: producers; variable used as weight: county).

|  | Coefficient | Std. Error | t-Ratio | *p*-Value |
|---|---|---|---|---|
| Products | 0.202004 | 0.0991651 | 2.037 | 0.1786 |
| Turnover | 0.0906022 | 0.587593 | 0.1542 | 0.8916 |
| Gross Profit | 0.624531 | 0.313168 | 1.994 | 0.1843 |
| Belsley–Kuh–Welsch collinearity diagnostics: | | | | |
| Lambda (eigenvalues of X'X, largest to smallest) | | | | |
| Lambda | Cond. index | Products | Turnover | Gross Profit |
| 2.422 | 1.000 | 0.033 | 0.067 | 0.034 |
| 0.454 | 2.310 | 0.064 | 0.932 | 0.073 |
| 0.124 | 4.412 | 0.903 | 0.001 | 0.894 |

**Statistics based on the weighted data:**

| | |
|---|---|
| The sum of squared residuals | 9.175181 |
| Uncentered R-squared | 0.870163 |
| F (3. 2) | 4.467958 |
| Log-likelihood | −8.612353 |
| Schwarz criterion | 22.05302 |

| The standard regression error | 2.141866 |
| Centered R-squared | 0.948827 |
| P-value (F) | 0.188291 |
| Akaike criterion | 23.22471 |
| Hannan-Quinn | 20.08002 |
| Mean dependent var | 3.000000 |
| The sum of squared residuals | 7.855149 |
| S.D. dependent var | 2.449490 |
| The standard regression error | 1.981811 |

**Variance Inflation Factors**

VIF ($j$) = 1/ (1 − $R$ ($j$)^2), where $R(j)$ is the multiple correlation coefficient between variable $j$ and the other independent variables.

The minimum possible value = 1

Values > 10 may indicate a collinearity problem

Products 1.144

Turnover 1.105

Gross Profit 1.046

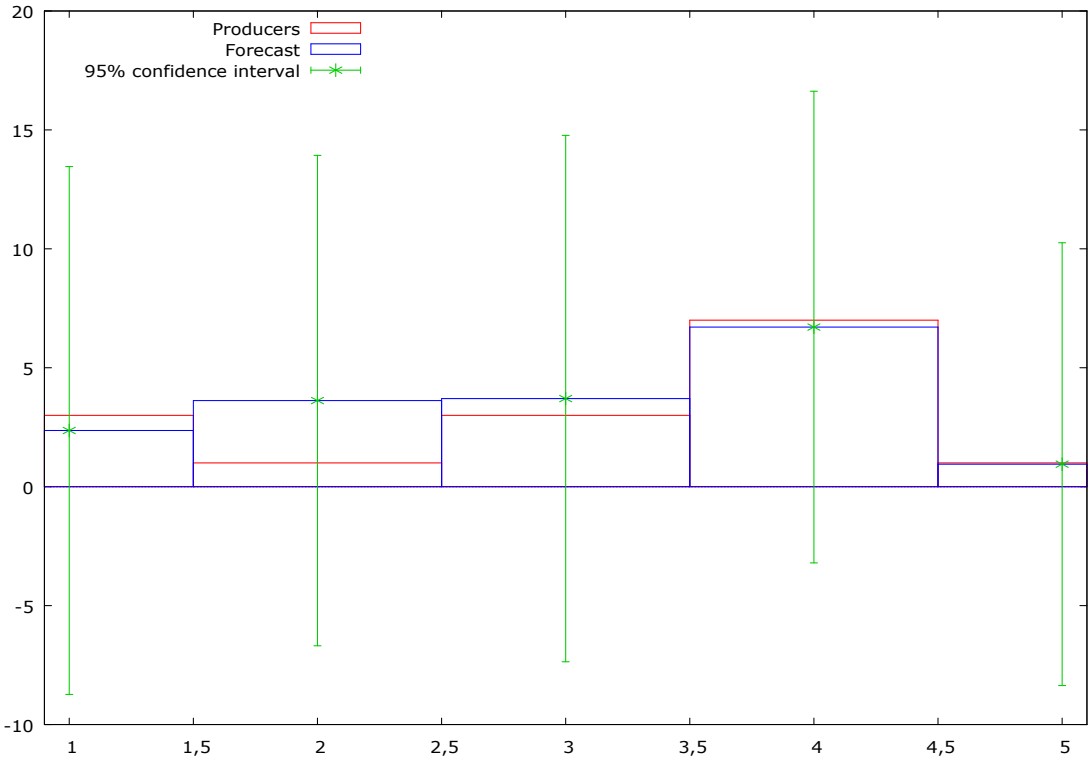

**Figure 5.** For 95% confidence intervals, *t* (2, 0.025) = 4.303.

Figure 5 shows the predicted distribution on the 95% confidence interval of the model-dependent variable, the distribution that reflects the homogeneity of the data and the validity of the model. This confirms the working hypothesis.

## 4. Results

The proposed ZML model may have an interdisciplinary applicability depending on its aims and the assigned performance criteria. In the present study, the use of the ZML model was carried out in order to calculate the geographical distribution of the certified traditional products and of the product categories to which they belong by calculating the concentration degree on the maximum of the statistical performance score (Table 6).

**Table 6.** The distribution of certified traditional products by category and by county with the identification of the concentration degree on the maximum of the statistical performance score.

| County | Product Categories ($j_m$) | Products ($i_k$) | $\left(\frac{i_k}{\sum_{k=0}^{p} i_k}\right)^2$ | $1 - \left(\frac{i_k}{\sum_{k=0}^{p} i_k}\right)^2$ |
|---|---|---|---|---|
| Alba | 5 | 53 | 0.081917 | 0.99329 |
| Arad | 3 | 3 | 0.004637 | 0.999979 |
| Arges | 4 | 37 | 0.057187 | 0.99673 |
| Bacau | 1 | 1 | 0.001546 | 0.999998 |
| Bihor | 2 | 6 | 0.009274 | 0.999914 |
| Bistrita Nasaud | 2 | 2 | 0.003091 | 0.99999 |
| Botosani | 5 | 29 | 0.044822 | 0.997991 |
| Brasov | 5 | 175 | 0.270479 | 0.926841 |
| Braila | 1 | 1 | 0.001546 | 0.999998 |
| Bucuresti | 1 | 4 | 0.006182 | 0.999962 |
| Buzau | 2 | 24 | 0.037094 | 0.998624 |
| Caras Severin | 1 | 3 | 0.004637 | 0.999979 |
| Cluj | 4 | 8 | 0.012365 | 0.999847 |
| Constanta | 1 | 1 | 0.001546 | 0.999998 |
| Covasna | 3 | 25 | 0.03864 | 0.998507 |
| Dambovita | 2 | 9 | 0.01391 | 0.999807 |
| Galati | 1 | 7 | 0.010819 | 0.999883 |
| Harghita | 1 | 3 | 0.004637 | 0.999979 |
| Hunedoara | 4 | 11 | 0.017002 | 0.999711 |
| Iasi | 3 | 12 | 0.018547 | 0.999656 |
| Ilfov | 2 | 13 | 0.020093 | 0.999596 |
| Maramures | 6 | 65 | 0.100464 | 0.989907 |
| Mures | 1 | 1 | 0.001546 | 0.999998 |
| Neamt | 2 | 36 | 0.055641 | 0.996904 |
| Olt | 1 | 2 | 0.003091 | 0.99999 |
| Prahova | 2 | 5 | 0.007728 | 0.99994 |
| Satu Mare | 5 | 27 | 0.041731 | 0.998259 |
| Salaj | 3 | 13 | 0.020093 | 0.999596 |
| Sibiu | 2 | 11 | 0.017002 | 0.999711 |
| Suceava | 4 | 18 | 0.027821 | 0.999226 |
| Timis | 1 | 4 | 0.006182 | 0.999962 |
| Tulcea | 3 | 19 | 0.029366 | 0.999138 |
| Vaslui | 1 | 2 | 0.003091 | 0.99999 |
| Valcea | 3 | 12 | 0.018547 | 0.999656 |
| Vrancea | 1 | 5 | 0.007728 | 0.99994 |
| $\sum \left(1 - \left(\frac{i_k}{\sum_{k=0}^{p} i_k}\right)^2\right)$ | | | | 5.993225 |
| $\text{ZML*} = \dfrac{\sum_{m=1}^{r} j_m}{\sqrt{\dfrac{\sum_{m=1}^{r} j_m \cdot \left[1 - \left(\frac{i_k}{\sum_{k=0}^{p} i_k}\right)^2\right]}{(\sum_{m=1}^{r} j_m) - 1}}}$ | | | | 5.839928 |
| $\lim_{j \to \infty}(|\max(i_k) - \text{ZML*}|)$ | | | | 6 |

Source: Author's calculations based on the information from [35].

Following the application of the model in Table 6, a concentration degree was calculated on the maximum statistical performance score for traditional products by category of certified products and by county of 5.84, corresponding to a number of 5–6 categories of certified products per county.

In order to demonstrate the viability of the model, we estimated the concentration of the certified traditional food product by county (Table 7).

**Table 7.** The certified traditional product concentration degree, calculated by category of certified traditional products and by county.

| No. of Counties | No. of Certified Traditional Product Categories/County | No. of Certified Traditional Products | No. of Products/Category of Certified Traditional Products | No. of Products/Category of Certified Traditional Products and by County |
|---|---|---|---|---|
| **12** | 1 | 34 | 34 | 3 |
| **8** | 2 | 106 | 53 | 7 |
| **6** | 3 | 84 | 28 | 5 |
| **4** | 4 | 74 | 19 | 5 |
| **4** | 5 * | 284 | 57 | 14 * |
| **1** | 6 * | 65 | 11 | 11 * |

Source: Author's calculations based on the information from [35].

Note: * = upper limit

Table 7 shows that the upper limit of the performance score. The score is characterized by the equation $\lim_{i \to \infty}(|\max(i_k) - ZML^*|)$. Eleven traditional products (11*) were certified in the six categories (6*) and were attributed with maximum performance. Because the ideal case hypothesis is difficult to put into practice, one may notice that the ZML* function identifies the performance score at a limit of 5.84 by including in the model counties that have five categories of certified traditional products, with 14 certified traditional products in each category.

The model was applied to dynamically calculate the evolution of the statistical performance score for the certified traditional products per year and per product category, as follows.

The calculation of the statistical performance score in Table 8 for 2014 indicated that the meat and meat products category was representative of the producers' options with 117 certified products out of a total of 296 products registered in the NTPR.

**Table 8.** The calculation of the statistical performance score for the traditional products registered in 2014 for each product category.

| 2014 | | | | |
|---|---|---|---|---|
| No. of Certified Traditional Product Categories | No. of products/Category of Certified Traditional Products | $\left(\dfrac{i_k}{\sum_{k=0}^{p} i_k}\right)$ | $\left(\dfrac{i_k}{\sum_{k=0}^{p} i_k}\right)^2$ | $1 - \left(\dfrac{i_k}{\sum_{k=0}^{p} i_k}\right)^2$ |
| Other | 0 | 0.00% | 0 | 1 |
| Beverages | 5 | 1.69% | 0.000285 | 0.999715 |
| Meat and meat products | 117 | 39.53% | 0.156239 | 0.843761 |
| Milk and milk products | 78 | 26.35% | 0.069439 | 0.930561 |
| Vegetables, fruit | 44 | 14.86% | 0.022096 | 0.977904 |
| Bread, bakery and pastry | 37 | 12.50% | 0.015625 | 0.984375 |
| Fish | 15 | 5.07% | 0.002568 | 0.997432 |
| **Total 2014** | **296** | 100.00% | 0.266253 | 6.733747 |
| $\sum\left(1 - \left(\dfrac{i_k}{\sum_{k=0}^{p} i_k}\right)^2\right)$ | | | | 2.842621 |
| $ZML^* = \dfrac{\sum_{m=1}^{r} jm}{\sqrt{\dfrac{\sum_{m=1}^{r} jm \cdot \left[1 - \left(\frac{i_k}{\sum_{k=0}^{p} i_k}\right)^2\right]}{(\sum_{m=1}^{r} jm) - 1}}}$ | | | | 104.1292 |
| $\lim_{j \to \infty}(|\max(i_k) - ZML^*|)$ | | | | 117 |

Source: Author's calculations based on the information from [35].

The performance score of 104.13 also allows for inclusion in the performance area of the milk and milk products category, which was the second option of producers in the preferred order of product certification (Figure 6).

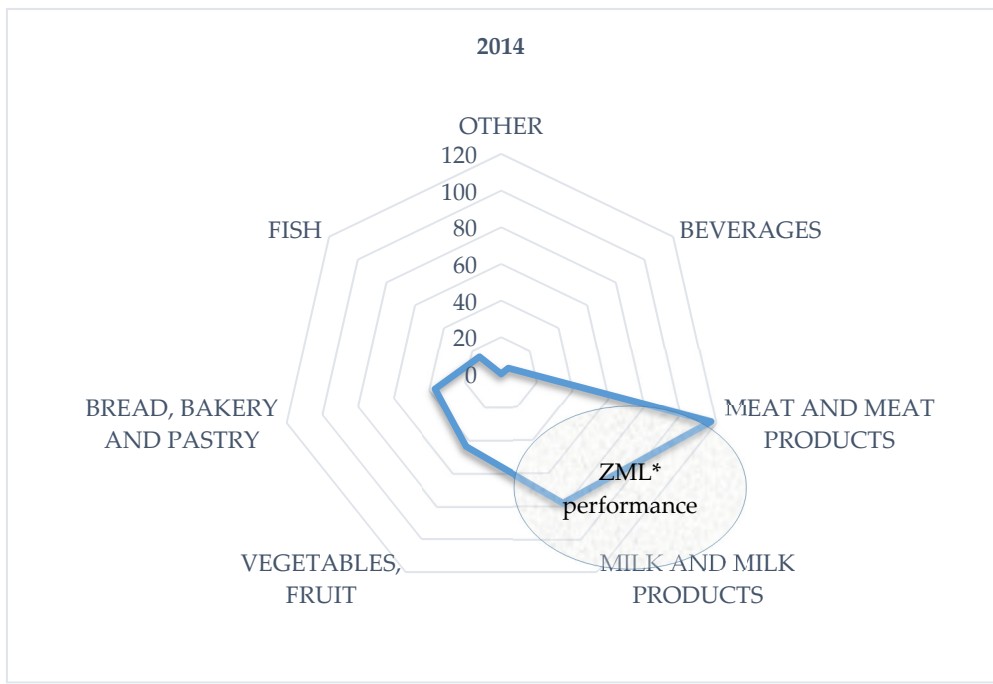

**Figure 6.** The graphical representation of certified traditional product statistical performance score calculation in 2014 per product category.

The calculation of the statistical performance score as shown in Table 8 for 2015 indicates that it was the same value for the meat and meat products category. This was representative of producers' choices with 79 registered products out of 203 total products registered in the NTPR (see Table 9).

**Table 9.** The calculation of the statistical performance score for the traditional products registered in 2015 per product category.

| **2015** | | | | |
|---|---|---|---|---|
| **No. of Certified Traditional Products** | **No. of Products/Category of Certified Traditional Products** | $\left(\dfrac{i_k}{\sum_{k=0}^{p} i_k}\right)$ | $\left(\dfrac{i_k}{\sum_{k=0}^{p} i_k}\right)^2$ | $1 - \left(\dfrac{i_k}{\sum_{k=0}^{p} i_k}\right)^2$ |
| Other | 2 | 0.99% | $9.71 \times 10^{-5}$ | 0.999903 |
| Beverages | 15 | 7.39% | 0.00546 | 0.99454 |
| Meat and meat products | 79 | 38.92% | 0.151447 | 0.848553 |
| Milk and milk products | 46 | 22.66% | 0.051348 | 0.948652 |
| Vegetables, fruit | 13 | 6.40% | 0.004101 | 0.995899 |
| Bread, bakery and pastry | 41 | 20.20% | 0.040792 | 0.959208 |
| Fish | 7 | 3.45% | 0.001189 | 0.998811 |
| **Total 2015** | **203** | 100.00% | 0.254435 | 6.745565 |
| $\sum\left(1 - \left(\dfrac{i_k}{\sum_{k=0}^{p} i_k}\right)^2\right)$ | | | | 2.805321 |
| $\text{ZML*} = \dfrac{\sum_{m=1}^{r} jm}{\sqrt{\dfrac{\sum_{m=1}^{r} jm \cdot \left[1 - \left(\dfrac{i_k}{\sum_{k=0}^{p} i_k}\right)^2\right]}{(\sum_{m=1}^{r} jm) - 1}}}$ | | | | 72.36248 |
| $\lim_{j \to \infty}(|\max(i_k) - \text{ZML*}|)$ | | | | 79 |

Source: Author's calculations based on the information from [35].

The performance score of 72.36 allows for inclusion in the performance area of the milk and milk products category. This was the producers' second choice in the order of preference with 46 products (Figure 7).

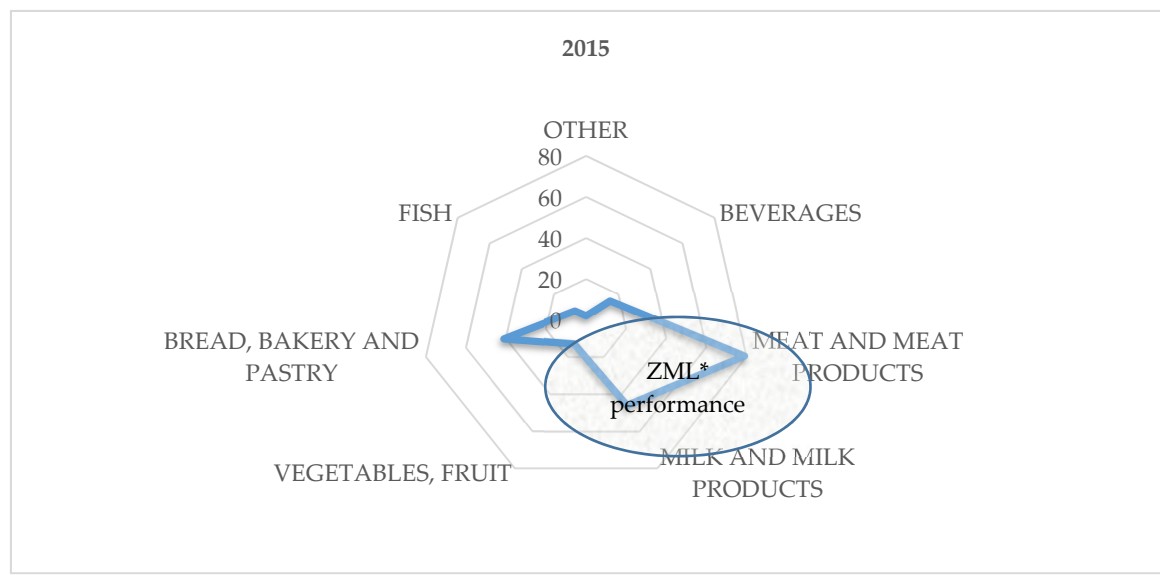

**Figure 7.** The graphical representation of the statistical performance score for the traditional products registered in 2015 per product category.

The calculation of the statistical performance score in Table 10 for 2016 indicates that the meat and meat products category was again representative of producers' choices with 25 certified products out of 55 total products registered in the NTPR.

**Table 10.** The calculation of the statistical performance score for the traditional products registered per product category in 2015.

| 2016 | | | | |
|---|---|---|---|---|
| **No. of Certified Traditional Products** | **No. of Products/Category of Certified Traditional Products** | $\left(\dfrac{i_k}{\sum_{k=0}^{p} i_k}\right)$ | $\left(\dfrac{i_k}{\sum_{k=0}^{p} i_k}\right)^2$ | $1-\left(\dfrac{i_k}{\sum_{k=0}^{p} i_k}\right)^2$ |
| Other | 0 | 0.00% | 0 | 1 |
| Meat and meat products | 25 | 45.45% | 0.206612 | 0.793388 |
| Milk and milk products | 1 | 1.82% | 0.000331 | 0.999669 |
| Vegetables, fruit | 21 | 38.18% | 0.145785 | 0.854215 |
| Bread, bakery and pastry | 6 | 10.91% | 0.011901 | 0.988099 |
| Fish | 2 | 3.64% | 0.001322 | 0.998678 |
| **Total 2016** | **55** | 100.00% | 0.36595 | 6.63405 |
| $\sum\left(1-\left(\dfrac{i_k}{\sum_{k=0}^{p} i_k}\right)^2\right)$ | | | | 2.879681 |
| $\text{ZML*} = \dfrac{\sum_{m=1}^{r} j_m}{\sqrt{\dfrac{\sum_{m=1}^{r} j_m \cdot \left[1-\left(\frac{i_k}{\sum_{k=0}^{p} i_k}\right)^2\right]}{(\sum_{m=1}^{r} j_m)-1}}}$ | | | | 19.09934 |
| $\lim_{j\to\infty}(|\max(i_k)-\text{ZML*}|)$ | | | | 25 |

Source: Author's calculations based on the information from [35].

The 19.09 performance score allows for full inclusion in the performance area of the vegetables and fruit category, which was the second choice of producers in the order of preference, with 21 products. The performance distribution was bipolar in this case (Figure 8). This is a case of the model where two categories simultaneously meet performance conditions. This case manifests itself in the event of low population density in which alternative options tend towards zero (other = 0; beverages = 0; milk and milk products = 1; fish = 2; bread, bakery, and pastry = 6).

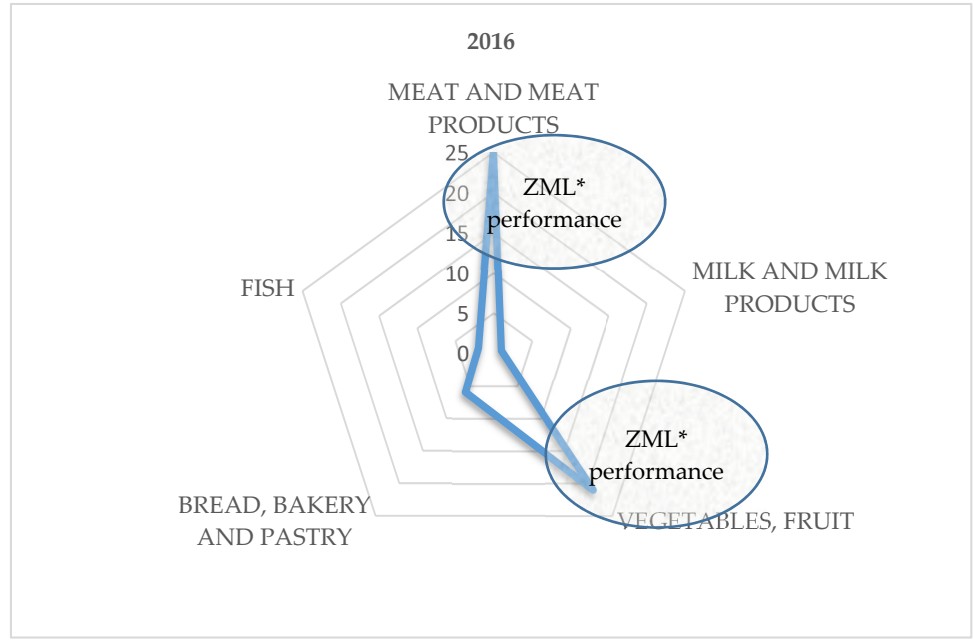

**Figure 8.** The graphical representation of the statistical performance score for traditional products registered in 2016 per product category.

The calculation of the statistical performance score in Table 11 for 2017 indicated that the milk and milk products category was representative of the producers' choices, with 11 certified products out of 36 total products registered in the NTPR. It is almost equal to the vegetables and fruit category for a 10 total products.

**Table 11.** The calculation of the statistical performance score for the traditional products registered per product category in 2017.

| **2017** | | | | |
|---|---|---|---|---|
| **No. of Certified Traditional Products** | **No. of Products/Category of Certified Traditional Products** | $\left(\dfrac{i_k}{\sum_{k=0}^{p} i_k}\right)$ | $\left(\dfrac{i_k}{\sum_{k=0}^{p} i_k}\right)^2$ | $1-\left(\dfrac{i_k}{\sum_{k=0}^{p} i_k}\right)^2$ |
| Other | 0 | 0.00% | 0 | 1 |
| Beverages | 1 | 2.78% | 0.000772 | 0.999228 |
| Meat and meat products | 9 | 25.00% | 0.0625 | 0.9375 |
| Milk and milk products | 11 | 30.56% | 0.093364 | 0.906636 |
| Vegetables, fruit | 10 | 27.78% | 0.07716 | 0.92284 |
| Bread, bakery and pastry | 5 | 13.89% | 0.01929 | 0.98071 |
| FISH | 0 | 0.00% | 0 | 1 |
| **Total 2017** | **36** | 100.00% | 0.253086 | 6.746914 |
| $\sum\left(1-\left(\dfrac{i_k}{\sum_{k=0}^{p} i_k}\right)^2\right)$ | | | | 2.904073 |
| $ZML^* = \dfrac{\sum_{m=1}^{r} j_m}{\sqrt{\dfrac{\sum_{m=1}^{r} jm \cdot \left[1-\left(\dfrac{i_k}{\sum_{k=0}^{p} i_k}\right)^2\right]}{(\sum_{m=1}^{r} jm)-1}}}$ | | | | 12.39638 |
| $\lim\limits_{i\to\infty}\left(\lvert \max(i_k) - ZML^* \rvert\right)$ | | | | 11 |

Source: Author's calculations based on the information from [35].

The 12.39 performance score allows for inclusion in the performance area of the meat and meat products category, which was the third option of producers in the order of preference with 9 products. The performance distribution was, in this case, a tripolar one (Figure 9). This is another case of the model where three categories simultaneously fulfill performance conditions. The case

manifests itself for a low number of population density and for a low number of certified traditional products categories when alternative options tend towards zero (other = 0; fish = 0; beverages = 1; bread, bakery and pastry = 5).

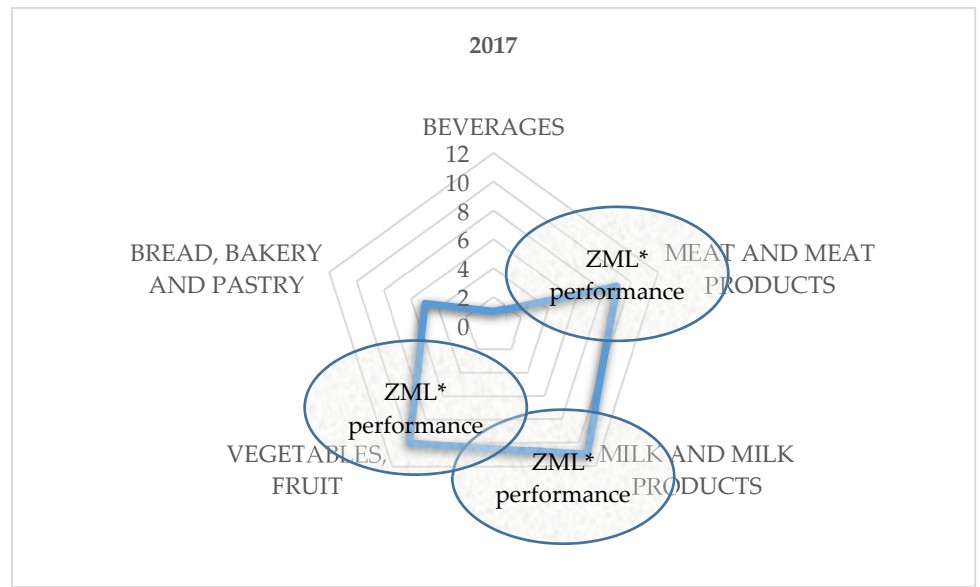

**Figure 9.** The graphical representation of the statistical performance score for the traditional products per product category registered in 2017.

The calculation of the statistical performance score in Table 12 for 2018 indicates that the vegetables and fruit and the bread, bakery, and pastry categories were representative of the producers' options, with 19 certified products from 57 total products registered in the NTPR.

**Table 12.** The calculation of the statistical performance score for the traditional products registered in 2018 per product category.

| **2018** | | | | |
|---|---|---|---|---|
| **No. of Certified Traditional Products** | **No. of Products/Category of Certified Traditional Products** | $\left(\dfrac{i_k}{\sum_{k=0}^{p} i_k}\right)$ | $\left(\dfrac{i_k}{\sum_{k=0}^{p} i_k}\right)^2$ | $1 - \left(\dfrac{i_k}{\sum_{k=0}^{p} i_k}\right)^2$ |
| Other | 0 | 0.00% | 0 | 1 |
| Meat and meat products | 11 | 19.30% | 0.037242 | 0.962758 |
| Milk and milk products | 5 | 8.77% | 0.007695 | 0.992305 |
| Vegetables, fruit | 19 | 33.33% | 0.111111 | 0.888889 |
| Bread, bakery and pastry | 19 | 33.33% | 0.111111 | 0.888889 |
| Fish | 3 | 5.26% | 0.00277 | 0.99723 |
| **Total 2018** | **57** | 100.00% | 0.269929 | 6.730071 |
| $\sum\left(1 - \left(\dfrac{i_k}{\sum_{k=0}^{p} i_k}\right)^2\right)$ | | | | 2.900446 |
| $ZML^* = \dfrac{\sum_{m=1}^{r} j_m}{\sqrt{\dfrac{\sum_{m=1}^{r} j_m \cdot \left[1 - \left(\frac{i_k}{\sum_{k=0}^{p} i_k}\right)^2\right]}{(\sum_{m=1}^{r} jm) - 1}}}$ | | | | 19.65215 |
| $\lim\limits_{i \to \infty}(|\max(i_k) - ZML^*|)$ | | | | 19 |

Sources: Author's calculations based on the information from [35].

The 19.45 performance score does not allow other product categories to be included in the performance area. The performance distribution is a bipolar one (Figure 10).

This is a case of the model where two categories simultaneously meet performance conditions. The case manifests itself in the presence of low population density in which alternative options tend towards zero (other = 0; beverages = 0; milk and milk products = 5; fish = 3; meat and meat products = 11).

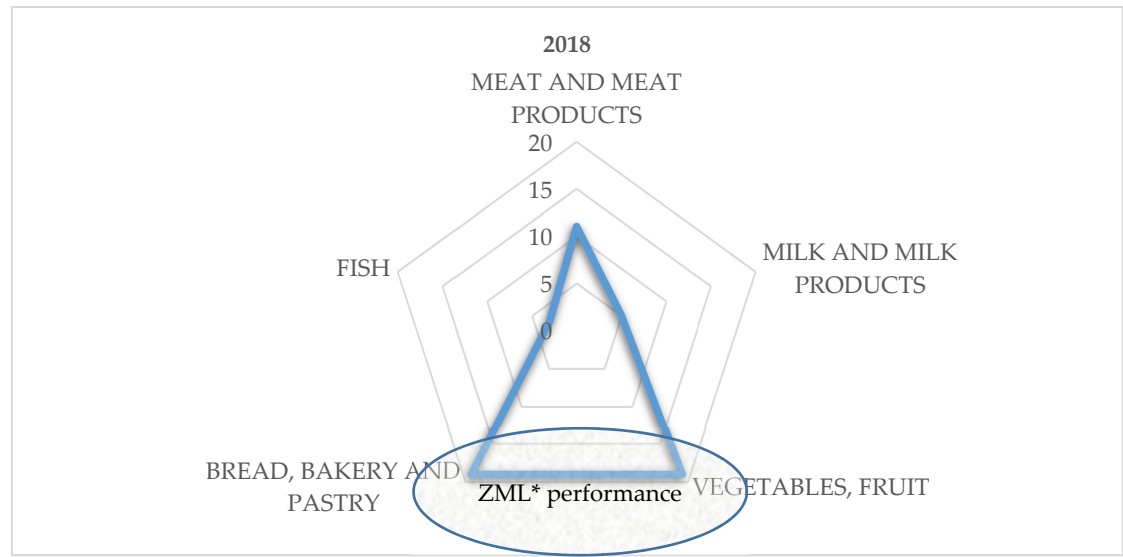

**Figure 10.** The graphical representation of the statistical performance score for the traditional products registered in 2018 per product category.

In order to quantify the economic effects in the context of Hypotheses 3, 4, and 5, we gathered the financial data from all producers who applied for registration in the NTPR in 2018. We compiled it with data collected on average over the last five years for a number of 15 entities, the remaining eight producers being newly established or authorized physical persons/sole traders or being organized under other forms of family organization. In their case, the data could not be statistically collected in relation to the number of products. Informations collected for 43 products out of 57 total products registered, meaning a 75% significance considered relevant to the proposed case study.

The information obtained from the centralization results of the fifteen producers generated data shown in Table 13.

**Table 13.** The calculation of the statistical performance score for the traditional products registered in 2018 per product category.

| Categories (5) | No. of Producers | % Representation Producer/Category | County | % Representation per Counties | No. of Certified Products | % of Certified Products |
|---|---|---|---|---|---|---|
| Meat and Meat Products | 3 | 20% | 2 | 20% | 4 | 9.30% |
| Milk and Milk Products | 1 | 6.67% | 1 | 10% | | 9.30% |
| Vegetables-Fruit | 3 | 20% | 2 | 20% | 16 | 37.21% |
| Bread, Bakery and Pastry | 7 | 46.67% | 6 | 60% | 16 | 37.21% |
| FISH | 1 | 6.67% | 1 | 10% | 3 | 6.98% |
| **Total** | 15 | 100% | 10 | 100% | 43 | 100% |
| Max | Bread, bakery and pastry | | Bread, bakery and pastry | | Bread, bakery and pastry Vegetables-fruit | |

| Categories (5) | Turnover (%) | Gross Profit/ Loss (%) | Income (%) | Total Capital (%) | Total Fixed Assets (%) | Total Current Assets (%) | Debts (%) |
|---|---|---|---|---|---|---|---|
| Meat and meat products | 272.58% | 209.36% | 218.91% | 176.49% | 815.77% | 305.34% | 560.74% |
| Milk and milk products | 119.27% | 433.02% | 86.36% | 2356.29% | 169.66% | 588.64% | 304.35% |
| Vegetables-fruit | 118.09% | 58.45% | 25.29% | 230.10% | 54.77% | 24.65% | 78.04% |

| Bread, bakery and pastry | 73.92% | 546.31% | 56.32% | 392.27% | 162.60% | 37.49% | 49.80% |
|---|---|---|---|---|---|---|---|
| Fish | 57.72% | 46.20% | 71.38% | 53.72% | 69.26% | 102.78% | 117.92% |
| **Total** | 124.43% | 340.45% | 85.64% | 425.05% | 265.92% | 129.59% | 179.15% |
| Max | Meat and meat products | Bread, bakery and pastry | Meat and meat products | Milk and milk products | Meat and meat products | Milk and milk products | Meat and meat products |

Source: Author's calculations based on the information from [35].

The statistical calculation of the indicator distributions presented in Table 13 based on the confidence interval for the median indicator shown in Figure 11 reveals the homogeneity of the data taken into account in the construction of the statistical model for assessing the capacity of the agricultural sector within the framework of the economic opportunities generated by promotion campaigns.

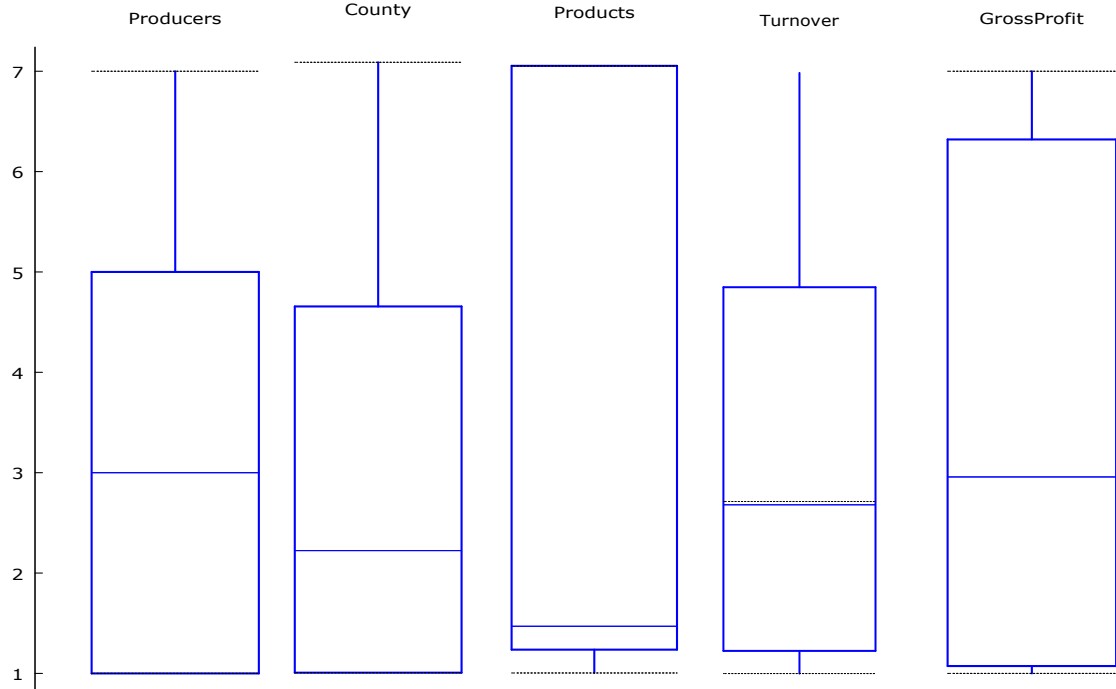

**Figure 11.** The statistical calculation of the main indicator distributions depending on the confidence interval for their median/center line; source: projection of the authors using the Gretl statistical program, version 2018.

From the point of view of the descriptive statistics, the indicators used in the model calculation are presented in Table 14.

**Table 14.** Descriptive statistics using Remarks 1–5 as the number of categories.

| Variable | Median | Maximum | Dev. Stand. | C.V. |
|---|---|---|---|---|
| Categories | 3.0000 | 5.0000 | 1.5811 | 0.52705 |
| Producers | 3.0000 | 7.0000 | 2.4495 | 0.81650 |
| County | 2.0000 | 6.0000 | 2.0736 | 0.86402 |
| Products | 4.0000 | 16.000 | 6.7676 | 0.78693 |
| Turnover | 1.1809 | 2.7258 | 0.85061 | 0.66290 |
| Gross Profit | 2.0936 | 5.4631 | 2.2405 | 0.86618 |
| Income | 0.71380 | 2.1891 | 0.74642 | 0.81441 |
| Equity | 2.3010 | 23.563 | 9.6612 | 1.5054 |
| Fixed assets | 1.6260 | 8.1577 | 3.1815 | 1.2505 |
| Current assets | 1.0278 | 5.8864 | 2.3877 | 1.1275 |
| Debts | 1.1792 | 5.6074 | 2.1374 | 0.96205 |

Source: Author's calculations based on the information from [35].

## 5. Discussion

The proposed model quantifies the degree of concentration in the productive and economic performance areas of traditional products. The certified traditional product and product categories dynamic analysis evolution and the identification of the annual performance score were used to visualize the score translation phenomenon between categories since the establishment of the NTPR, which depended on circumstantial factors and economic opportunities generated by the traditional product promotion campaigns. The conceptualized model shows a significant improvement in comparison to an earlier, similar pattern [19] in terms of the identification of the performance score and the difference from the leader of any element from the analyzed population. Moreover, it highlights the additional elements that are necessary to assess the relative frequency distribution and quantification of the impact of the concentration on the entire analyzed sample.

Based on the data taken from the NTPR, this model allowed for the calculation of the dynamic performance areas based on traditional product categories most often used by manufacturers for registration of the bipolar and the tripolar performance cases during the period between 2016 and 2017. According to this model, the validity of the first and second hypotheses was demonstrated.

To demonstrate the other hypotheses, a cumulative model based on the least weighted squares was developed. It generated 87% validity, statistically meaningful results, and a 95% significance value corresponding to a centered R-squared. Hypotheses 3–5 were demonstrated as follows:

- In 2018, 46.67% of the traditional product manufacturers (15) focused mainly on the bakery and pastry sector.
- From the point of view of regional distribution, the bread, bakery, and pastry category was represented in 6 counties out of the total of 10 counties that applied for certifications of the traditional products at the sample level (60%—category representation in all the counties).
- From the point of view of the number of certified products (43 products), out of a total of 57 for the year 2018, the distribution is bipolar. The impact categories were bread, bakery, and pastry and vegetables and fruit, with 37.21% for each category.
- From the point of view of maximizing sales, as assumed in Hypothesis 3, meat and meat products, with the highest number of entries in the Register, show returns of over 200% at the level of financial indicators of turnover, gross profit, and revenues. This certifies the sustainable development of the branch that has growth rates for all seven economic indicators, which were part of the analysis.
- From the gross profitability point of view (the third hypothesis), it was noticed that the statistical performance score indicates, for the bread, bakery, and pastry category, the direct interest of the producers. This category reached the maximum level, in terms of the number of producers, the traditional representativeness, and the number of products registered during this period. The fruit and vegetable category of products was represented by the constant interest of the producers, manifested by the most flattened evolution curve regarding the dynamics of the number of certified products, with a special note that the seasonal impact and external factors are those factors that have a direct impact on the reduced profitability in the sector. In this context, the limited export capacity, the annual production, and the direct and fierce competition with top international products cover the same consumption mentioned.
- From the capitalization point of view (the fourth hypothesis), the category with the highest accumulation rate of current assets (i.e., milk and milk products) is also the category with the highest accumulation of capital, and the other condition is fulfilled (i.e., the difference between profitability and the increase in turnover, in the case where this category is over 300%).
- In line with the fifth hypothesis, the most technologically demanding category is meat and meat products, which has a direct impact on capital accumulation; the return represents only 176% and has an impact on debt accumulation with borrowed capital, which is twice as high as for milk and milk products. A debt growth rate, such as that for bread, bakery, and pastry, always involves higher returns in terms of entity capitalization of equity accumulation.

- From the non-performance point of view of, one can notice that fish products do not benefit from a significant contribution to the economic opportunities generated by the promotion campaigns. When non-performance is combined with the incipient stage of aquaculture policy implementation, they represent the last desirable option of the entities. The number of certified products (3 and 6.98%, respectively) in a non-performance sector generated a negative growth rate (sub unitary) of the turnover, gross profit, and entity capitalization.

The economic development concentration analysis allowed for the identification of performance areas in the traditional product sector based on the ZML statistical model, demonstrating that the development of this branch is directly proportional to the national and international recognition of cultural and historical values. However, the development is also related to the market saturation level with products included in the category of traditional product.

The sustainable development of the branch aims at assessing the economic performance identified by means of both the statistical performance score, added to the financial components, and by means of the growth rates by category of certified products. Their economic returns known to have a perpetually positive dynamic for economic indicators (turnover, gross profit, total income, equity, fixed assets, current assets, and total debts). The performed analysis prospectively assessed the dynamics of the certified traditional product market in Romania. With the highest performance score, the category of meat and meat products is characterized by the need for high technology assets. The milk and milk product category reaches the maximum level from all analyzed categories of traditional products in terms of capitalization, accumulation of current assets and profitability.

On the other hand, concentration degree analysis shows the poor sustainable development of the traditional fish product category. In addition to its negative returns, it also shows a dynamic high debt build-up degree by proving the need for support of national and European bodies, both through regulatory and financial measures, contributing to the fruition of an extensive productive area in Romania and to socio-cultural traditions related to the consumption of fish in the Danube Delta area.

## 6. Conclusions

We analyzed the dynamics of the sustainable evolution of the traditional product market in Romania using data for the period of operation of the NTPR.

The research shows that there is a dependency relationship between the development of trade with certain categories of traditional products and the sustainability induced by the financial profitability.

The study also shows (by absolute novelty method) that the performance of the cloud identifies and the gap between the leader and any element of the analyzed population.

The ZML model identified cases of bipolar and tripolar performance. All five working hypotheses have been tested and validated to demonstrate the model. The model indicates that sustainable development focuses on products whose processing requires high fixed asset technologies, high profitability, and high capitalization. In the opinion of the authors, traditional products can be addressed in a sustainable way, taking into account all the impact elements presented in the paper. Managers can offer new perspectives and profound business analyses.

These elements involve the harmonization of regional values with European sustainability values. This study has intrinsic value because it presents a model of concentration and extrinsic value, analyzing the sustainable development of the traditional product industry in Romania. The results of the study can be use by interested managers but also by national authorities in the process of harmonizing with European values in terms of sustainability.

The research highlights elements of sustainable development of traditional agri-food production in Romania. Although there are positive aspects, the development of traditional production is still in its infancy, and there are large differences between product categories made by small producers. The concentration of the traditional product market with a low number of vegetable products is another characteristic of the Romanian market. The research is useful for economic agents, for the scientific environment, and for the implementation of national agro-food development strategies. The study limitations are about the regional approach of sustainable development, the analysis of a limited

number of indicators, and a limited number of years (due to the short period of existence of the NTPR).

**Author Contributions:** Conceptualization: S.S., V.M.A., and C.I.B.; methodology: M.L.Z.; formal analysis: V.M.A. and S.S.; resources: C.I.B.; writing—original draft preparation: M.L.Z. and V.M.A.

**Funding:** This research received no external funding.

**Conflicts of Interest:** The authors declare no conflict of interest.

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
