# Peer review of "The Development Analysis of the Romanian Traditional Product Market Based on the Performance Model for Sustainable Economic Development"

_sustainability, doi:10.3390/su11041123_

Round 1

Reviewer 1 Report

17 January, 2019

Review for Sustainability

Manuscript ID:  422903

Title: The Sustainable Development Analyse of Romanian Traditional Products Market based on Statistical Performance Model

General Comments:

The manuscript is focused on the economic development of traditional products in Romania.  This reviewer strongly believes the growth of this sector is critically important for producers and the public in Romania.  As written, this manuscript is very hard to read, to discern the objectives or aims, to have certainty of the methods, or to understand the results.  There were many misspelled words and sentence structure or writing technique resulted in my inability to understand or to assume the point of the sentence.  As well as I could understand this manuscript is focused only on the economic side of sustainability, which is critical, however that should be better reflected in the title and throughout the manuscript.  The Introduction begins with a heavy focus on the environmental side of sustainability.  I could not find any analysis of the environment in methods or results.  This is important work.  With rewriting for added clarity, strategic and concise Introduction, and concise description of the methods and results, this manuscript could be published. 

Introduction is much better written than the abstract yet it does not come to the point and does not clearly state the objectives.

This reviewer recommends rewriting the Introduction with more focus on economics or markets.  I was looking for factors in the model that accounted for the environment and food quality after reading the Introduction. 

Again, sentence structure lead me to skip several sentences because I could not decide what was be said.

Specific comments:

Title

Perhaps for the title:  Development Analyse of Romanian Traditional Products Market based on Statistical Performance Model for Sustainable economic development.

Abstract

Aims and objectives are considered to be the same.  Please organize objects and aims and use only one of the terminologies.

Many misspelled words and sentence structure greatly reduces clarity.  I was often guessing as to the point being made and tried to rewrite.  Example 1- Aims to quantify the degree of concentration in the productive and economic performance areas of traditional products.  I believe this is what the authors were trying to say but I am unsure.  Or the concentration of what?

Example 2 – line 28, The methods used are the domain-specific market national and international context research.  Please clarify.

 Example 3 – line 31, “the own statistical model.  I believe the authors were saying they developed a model – ZML (define ZML here) as well as in the Methods section.  Or could have written:  a model developed by the authors? 

I am not sure what a brench is.

Please describe the predicted tendencies according to ZML and what the harmonization is.  That is you suggest a solution but do not describe it.  Back it up with statistics please.

Introduction

This reviewer recommends rewriting the Introduction with more focus on economics or markets.  Some focus on the environment as related to public well-being or food security is important.  As is, authors’ introduction begins with a critique on past agriculture management and the environment.  Please relate that to manuscripts’ focus on opening new markets relevant to Romania’s cultures.  I was looking for factors in the model that accounted for the environment and food quality after reading the Introduction. 

Your paragraph (below) is at the heart of what I believe you are trying to measure.  What you are measuring is only a piece of the “evolution”.  Please describe further how it is important in your Introduction.

 “The evolution in achieving sustainable development objectives in Romania is lower than the EU 80 average, however, Romania is able to reduce the gaps in the sustainable development segment, only 81 that indicators such as the risk of poverty after social transfers as well as the risk of time-lag, makes 82 the process more difficult [14]. The environmental crisis trigger factors can be limited in the context 83 of implementing an action model involving: rational resource management, the implementation of 84 recycling technologies, pro-active economic processes in relation to environmental policy, the 85 support and development of green industries, and other measures regarding sustainable 86 development [15].

Is there a measure of food quality in you analysis?

Is this an objective scientific manuscript?  Line 45, techniques that produced irreversible effects in terms of climate change, soil erosion and biodiversity 45 loss [1]    Some would say one can improve soil health.  Is there any hope?

Please consolidate Introduction and add clear and concise objectives or aims.

Methods

Again sentence structure or writing leaves me with poor understanding.  Please describe first overall structure of the study.

Who did you interview?  I believe I see some pieces of the flowchart you describe in the Introductions.  Perhaps a flow chart in Methods would be more descriptive and clear.

Model is at the heart of the study but first one must understand the study.  Did you interview different persons and numbers every year?

Tables and graphs: 

Tables and graphs should stand alone.  I tried to figure out what you were saying in your methods and results by looking at the figures, tables and graphs.  I was again uncertain who you interviewed and what was analyzed.  Please clearly describe the structure of the study and objectivess. 

Please better organize tables and add more information in captions of tables figures and graphs.

Conclusions: 

It is unclear what you found.  Please rewrite conclusions to summarize your results and identify your most important findings.

Author Response

Dear Sir,

Thank you for your valuable observations, which helped improve the article. We attach the attached manuscript with the requested changes and answers to your observations.

Review for Sustainability

Manuscript ID:  422903 

General Comments:

The manuscript is focused on the economic development of traditional products in Romania.  This reviewer strongly believes the growth of this sector is critically important for producers and the public in Romania.  As written, this manuscript is very hard to read, to discern the objectives or aims, to have certainty of the methods, or to understand the results.  There were many misspelled words and sentence structure or writing technique resulted in my inability to understand or to assume the point of the sentence.  As well as I could understand this manuscript is focused only on the economic side of sustainability, which is critical, however that should be better reflected in the title and throughout the manuscript.  The Introduction begins with a heavy focus on the environmental side of sustainability.  I could not find any analysis of the environment in methods or results.  This is important work.  With rewriting for added clarity, strategic and concise Introduction, and concise description of the methods and results, this manuscript could be published. 

Introduction is much better written than the abstract yet it does not come to the point and does not clearly state the objectives.

This reviewer recommends rewriting the Introduction with more focus on economics or markets.  I was looking for factors in the model that accounted for the environment and food quality after reading the Introduction. 

Again, sentence structure lead me to skip several sentences because I could not decide what was be said.

R: Referring to gereral comments we filled in with relevant information  to all sections: introduction, review literature, research methodology, results and discussions and conclusions.

Specific comments:

Title

Perhaps for the title:  Development Analyse of Romanian Traditional Products Market based on Statistical Performance Model for Sustainable economic development.

R: We must point out that according to your recommendations we have modified the title from ”The Sustainable Development Analyse of Romanian Traditional Products Market based on Statistical Performance Model”  to ”The development Analysis  of the Romanian Traditional Products Market based on the  Performance Model for Sustainable Economic Development”.

Abstract

Aims and objectives are considered to be the same.  Please organize objects and aims and use only one of the terminologies.

R: We reorganized and reformulated the whole abstract; we have redefined our goals using unitary terminology (see lines 17-35).

Many misspelled words and sentence structure greatly reduces clarity.  I was often guessing as to the point being made and tried to rewrite.  Example 1- Aims to quantify the degree of concentration in the productive and economic performance areas of traditional products.  I believe this is what the authors were trying to say but I am unsure.  Or the concentration of what?

R example 1: Regarding English language and style, we want to specify that we used the professional services of a native speaker, the work being thoroughly revised. Therefore, the ambiguities of the concepts presented in the study were eliminated such as your example (see lines 17-19).

Example 2 – line 28, The methods used are the domain-specific market national and international context research.  Please clarify.

R example 2: Additional references have been added to the methods used both in abstract (see lines 31-35) and in the material and method section (see lines 364-383) for clarification purposes. Also, the syntax: The methods used are the domain-specific market national and international context research.” was deleted.

 Example 3 – line 31, “the own statistical model.  I believe the authors were saying they developed a model – ZML (define ZML here) as well as in the Methods section.  Or could have written:  a model developed by the authors? 

R example 3: We reformulated the phrase that contained the expression "the own statistical model"(see lines 31-33).

I am not sure what a brench is.

R: The expression ”brench”  was replaced with ”branch” (see lines 46; 356; 361)

Please describe the predicted tendencies according to ZML and what the harmonization is.  That is you suggest a solution but do not describe it.  Back it up with statistics please.

R: We have removed from the abstract the ambiguities regarding the predictive trends of the ZML model, and the results of the application of this model have been described in the conclusions section (see lines 682-701), a  preamble of them is also presented in the introduction (see lines 239-249).

Introduction

This reviewer recommends rewriting the Introduction with more focus on economics or markets.  Some focus on the environment as related to public well-being or food security is important.  As is, authors’ introduction begins with a critique on past agriculture management and the environment.  Please relate that to manuscripts’ focus on opening new markets relevant to Romania’s cultures.  I was looking for factors in the model that accounted for the environment and food quality after reading the Introduction. 

Your paragraph (below) is at the heart of what I believe you are trying to measure.  What you are measuring is only a piece of the “evolution”.  Please describe further how it is important in your Introduction.

 “The evolution in achieving sustainable development objectives in Romania is lower than the EU 80 average, however, Romania is able to reduce the gaps in the sustainable development segment, only 81 that indicators such as the risk of poverty after social transfers as well as the risk of time-lag, makes 82 the process more difficult [14]. The environmental crisis trigger factors can be limited in the context 83 of implementing an action model involving: rational resource management, the implementation of 84 recycling technologies, pro-active economic processes in relation to environmental policy, the 85 support and development of green industries, and other measures regarding sustainable 86 development [15].

Is there a measure of food quality in you analysis?

Is this an objective scientific manuscript?  Line 45, techniques that produced irreversible effects in terms of climate change, soil erosion and biodiversity 45 loss [1]    Some would say one can improve soil health.  Is there any hope?

Please consolidate Introduction and add clear and concise objectives or aims.

R: We want to specify that the information contained in the introduction section has been revised and reformulated. Also, we carefully reviewed the cited publications and introduced 50% more bibliographic references than there were originally (39 versus 24 mainly new and very new references) that capture the current trends of international research on the field of study (see lines 740-741; 764-792; 798-801). We also clarified the objectives (see lines 17-23) and rewritten the introduction according to the observations made by both reviewers, and removed the ambiguities on environmental factors that do not influence the research on the sustainable nature of the markets for traditional products, as  it was initially approached in the article (see lines 34-35).

We have pointed in introduction the quantitative aspects on measuring food quality and safety as presented by the Global Food Security Index (see lines 78-98).

Methods

Again sentence structure or writing leaves me with poor understanding.  Please describe first overall structure of the study.

Who did you interview?  I believe I see some pieces of the flowchart you describe in the Introductions.  Perhaps a flow chart in Methods would be more descriptive and clear.

Model is at the heart of the study but first one must understand the study.  Did you interview different persons and numbers every year?

R: We have reconsidered the methodology of research, meaning that we have included hypotheses (see lines 344-363) and we have provided additional information on the new statistical model proposed (see lines 370-383). We introduced, among other things, a survey scheme (see lines 364-369, figure 4). We have explained the methods used that do not include the interview but only database collections/queries and consultations of the national register for traditional products available online for each studied year. Hopefully, through additions and reinterpretations to the section on research methodology, we have achieved your requirements.

Tables and graphs: 

Tables and graphs should stand alone.  I tried to figure out what you were saying in your methods and results by looking at the figures, tables and graphs.  I was again uncertain who you interviewed and what was analyzed.  Please clearly describe the structure of the study and objectivess. 
Please better organize tables and add more information in captions of tables figures and graphs.

R: With regard to tables and charts, these were reorganized according to your indications and the other reviewer by dividing the section into two distinct parts. Also, on the recommendation of the other reviewer, we have moved tables 2-5 which become tables 1-4 (see lines 284; 306; 319; 331) and figures 3-5 which become figures 1-3 (see lines 298; 314; 339) in a new section (section 2).

Conclusions: 

It is unclear what you found.  Please rewrite conclusions to summarize your results and identify your most important findings.

R: We have completely rewritten, according to the indications received, the conclusions section pointing out the achievements obtained by writing the article in terms of performance of the traditional products market, the sustainable development of the branch and the tendencies of harmonization with the values of Europe. Another approach to the conclusions aimed to enhance the usefulness and limits of the new own model developed in the research (see lines 674-699; see lines 699-701).

R: Thank you for all your recommendations and suggestions and we consider that you have brought a great addition of value to our paper work. We hope that through the improvements made to the research we are in your assent.

Reviewer 2 Report

Major comments

1.     The Introduction should be considerably improved in terms of writing and exposure of ideas, where the current state of the research field should be reviewed carefully and key publications cited. Perhaps adding some new papers would be more beneficial for the readers because of the insufficiently theoretical part. The Introduction does not define the purpose of the work and its significance. Please highlight the novelty of your paper.

2.     My second major objection is that the methodology is not described thoroughly and is incomplete. Please refine the methodology.

3.     Section 3 Results is a mixture between Introduction, Legislative Framework and Facts regarding the geographical distribution of the traditional products 
based on descriptive statistics. The authors should revise the Section 3: should present only the results of applied methodology of previous Section Material and method, respectively should provide a concise and precise description of the results, their interpretation as well as the experimental conclusions that can be drawn; moving the objective (lines 230-234) in Introduction; it is recommended to present the facts of tables 2-5 and figures 3-5 in a separate section preferable before Section 2.

Minor comments

1. It is highly recommended for non-native English speakers (being one myself) to use the professional services of language editing. There are several typos and grammatical errors:

a)     On line 55 “caracteristics makes ” should be “characteristics make” 
. You should rather use "Those characteristics distinguish them from..." or a similar sentence for “Those caracteristics 
makes them different from 
…”

b)     On line 62 “has analyse “ should be “has analyzed” or “analyzed”

c)     On line 63 “have establish”  
should be “have established” or “established”. You should rather use "...finds evidence in favour of..." or a similar sentence for “The experts have establish that 
...”

d)     On line 65 “has transformed “ should be “was transformed”.  You should rather use " The Romanian 
agricultural model was transformed into a much more 
productive model under the impact of EU agricultural policies and due to the financial opportunities offered by the European Union funding 
 programs in the agricultural sector 
” than “The Romanian 
agricultural model has transformed under the impact of EU agricultural policies into a much more 
productive model, also due to the financial opportunities offered by the European Union funding 
 programs in the agricultural sector “

e)     On line 68 “a gastronomic” should be replaced by “the gastronomic”

f)     On line 69 “additinal “ should be “additional”.

g)     On line 81, the sentence should stop after “average”. A new phrase starts with “However,…”

h)    On line 117 “know” should be  “known”

i)      On lines 120-121 “Also, internationally know are the following: …” should be reformulated to remove the oral style

2. It is not very clear whether the ZML model is proposed, in this paper or has been developed through previous research. If the ZML is developed by Monica Laura Zlati 
(line 157) in this paper and line 185 where “we” mean all authors then it contrasts with Author Contribution (line 530) where only Monica Laura Zlati and Cezar Ionuţ Bichescu appear as contributors to the methodology. If the ZML model has been developed through previous research, the publication should be cited. 

3. There are too few references. For a paper of 22 pages in the main text, 24 references are far too few. You should select the key / essential papers most relevant for your study. I would suggest an increase in the number of references by at least half, or doubling. Also more accurate on citations.

Author Response

Dear Sir,

Thank you for your valuable observations, which helped improve the article. We attach the attached manuscript with the requested changes and answers to your observations.

Major comments

1.     The Introduction should be considerably improved in terms of writing and exposure of ideas, where the current state of the research field should be reviewed carefully and key publications cited. Perhaps adding some new papers would be more beneficial for the readers because of the insufficiently theoretical part. The Introduction does not define the purpose of the work and its significance. Please highlight the novelty of your paper.

R: With reference to major comments (observation 1) we want to specify that the information contained in the introduction section has been revised and reformulated. Also, we carefully reviewed the cited publications and introduced 50% more bibliographic references than there were originally (39 versus 24 mainly new and very new references) that capture the current trends of international research on the field of study (see lines 740- 741; 764-792; 798-801). We have redefined and added both in the abstract section (see lines 17-35) and introduction (see lines 83-98; 135-209; 225-249) comments on the novelty character, purpose, significance of the article and aspects related to the methodology of the research. Furthermore, in the conclusions we highlighted the intrinsic and extrinsic contribution of the authors in the field of interest, emphasizing both the continuity of research and the conceptualization of the model and its role in identifying the sustainable development of the researched sector (see lines 674-701).

2.     My second major objection is that the methodology is not described thoroughly and is incomplete. Please refine the methodology.

R: Regarding observation 2, we have reconsidered the methodology of research, meaning that we have included hypotheses (see lines 344-363) and we have provided additional information on the new statistical model proposed (see lines 370-383). We introduced, among other things, a survey scheme (see lines 364-369, figure 4). Hopefully, through additions and reinterpretations to the section on research methodology, we have achieved your requirements.

3.     Section 3 Results is a mixture between Introduction, Legislative Framework and Facts regarding the geographical distribution of the traditional products based on descriptive statistics. The authors should revise the Section 3: should present only the results of applied methodology of previous Section Material and method, respectively should provide a concise and precise description of the results, their interpretation as well as the experimental conclusions that can be drawn; moving the objective (lines 230-234) in Introduction; it is recommended to present the facts of tables 2-5 (and figures 3-5 in a separate section preferable before Section 2.

R: Regarding Section 3, according to your indications we moved the information on the objectives, the Legislative Framework and the geographical distribution of traditional products based on descriptive statistics, in a new section before the methodology (section 2- see lines 250-342). At the same time, on your recommendation, we have moved tables 2-5 which become tables 1-4 (see lines 284; 306; 319; 331) and figures 3-5 which become figures 1-3 (see lines 298; 314; 339) in section 2.

Minor comments

1.                  It is highly recommended for non-native English speakers (being one myself) to use the professional services of language editing. There are several typos and grammatical errors:

R 1: Regarding English language and style, we want to specify that we have used the professional services of a native speaker, the work being thoroughly revised.

R: Regarding the minor recommendations and suggestions, we report that we have used the services of a native English and all the work has been reviewed.

a)     On line 55 “caracteristics makes ” should be “characteristics make” . You should rather use "Those characteristics distinguish them from..." or a similar sentence for “Those caracteristics makes them different from …”

R a): We have reformulated (see lines 58-59)

b)     On line 62 “has analyse “ should be “has analyzed” or “analyzed”

R b): We have corrected (see line 64)

c)     On line 63 “have establish”  should be “have established” or “established”. You should rather use "...finds evidence in favour of..." or a similar sentence for “The experts have establish that ...”

R c): We have reformulated the entire phrase (see line 65-66)

d)     On line 65 “has transformed “ should be “was transformed”.  You should rather use " The Romanian agricultural model was transformed into a much more productive model under the impact of EU agricultural policies and due to the financial opportunities offered by the European Union funding programs in the agricultural sector” than “The Romanian agricultural model has transformed under the impact of EU agricultural policies into a much more productive model, also due to the financial opportunities offered by the European Union funding programs in the agricultural sector “

R d): We have reformulated the phrase after your recommendations (see lines 68-71)

e)     On line 68 “a gastronomic” should be replaced by “the gastronomic”

R e):  We reformatted the entire phrase (see lines 99-102)

f)     On line 69 “additinal “ should be “additional”.

R f): We reformatted the entire phrase (see lines 99-102)

g)     On line 81, the sentence should stop after “average”. A new phrase starts with “However,…”

R g): We reformatted the entire phrase (see lines 71-72)

h)    On line 117 “know” should be  “known”

R h): As far as the exposure of the ideas in English is concerned, the entire article has been reviewed by a native speaker of English, so this phrase containing a misspelled word has been removed from the article.

i)         On lines 120-121 “Also, internationally know are the following: …” should be reformulated to remove the oral style

R i): Due to the fact that certain phrases were reformulated, the blurring regarding the concepts presented in the study as the example given by you were removed from article.

2. It is not very clear whether the ZML model is proposed, in this paper or has been developed through previous research. If the ZML is developed by Monica Laura Zlati (line 157) in this paper and line 185 where “we” mean all authors then it contrasts with Author Contribution (line 530) where only Monica Laura Zlati and Cezar Ionuţ Bichescu appear as contributors to the methodology. If the ZML model has been developed through previous research, the publication should be cited. 

R 2: The ZML model is for the first time proposed in this paper (not developed by previous research). This is only developed by Zlati Monica Laura (we also rectified the author contributions section (see line 703).

3. There are too few references. For a paper of 22 pages in the main text, 24 references are far too few. You should select the key / essential papers most relevant for your study. I would suggest an increase in the number of references by at least half, or doubling. Also more accurate on citations.

R 3: The bibliographic references have been extended and updated taking into account the fact that other works with significant results have been added.

R: Thank you for all your recommendations and suggestions and we consider that you have brought a great addition of value to our paper work. We hope that through the improvements made to the research we are in your assent.

Round 2

Reviewer 2 Report

Thanks for your great effort to rework the paper. We can see that you have made changes and further analyzes that have added the benefits of your paper. I recommended to publish of your paper as it is.